

# Assimilation of passive microwave AMSR-2 satellite observations in a snowpack evolution model over North-Eastern Canada

Fanny Larue[1,2,3], Alain Royer[1,2], Danielle De Sève[3], Alexandre Roy[1,2,4], Emmanuel Cosme[5]

1 CARTEL, Université de Sherbrooke, Québec, Canada
5 2 Centre d'Études Nordiques, Québec, Canada
3 IREQ, Hydro-Québec, Québec, Canada
4 Département de Géographie, Université de Montréal, Québec, Canada
5 Institut des Géosciences de l'Environnement, IGE, UGA-CNRS, Grenoble, France

**Corresponding author:** Fanny Larue (fanny.larue@usherbrooke.ca)

**Abstract.** Over northeastern Canada, the amount of water stored in a snowpack, estimated by its snow water equivalent (SWE) amount, is a key variable for hydrological applications. The limited number of weather stations driving snowpack models over large and remote northern areas generates great uncertainty in SWE evolution. A data assimilation (DA) scheme was developed to improve SWE estimates by updating meteorological forcing data and snowpack states using passive microwave (PMW) satellite observations without using any surface-based data. In this DA experiment, a particle filter with a Sampled Importance Resampled algorithm (SIR) was applied and an inflation technique of the observation error matrix was developed to avoid ensemble degeneracy. The Advanced Microwave Scanning Radiometer – 2 (AMSR-2) brightness temperatures ($T_B$) observations were assimilated into a chain of models composed of the Crocus multi-layer snowpack model and radiative transfer models. The microwave snow emission model (Dense Media Radiative Transfer – Multi-Layers (DMRT-ML)), the vegetation transmissivity model ($\omega$-$\tau_{opt}$), and atmospheric and soil radiative transfer models were calibrated to simulate the contributions from the snowpack, the vegetation and the soil, respectively, at the top of the atmosphere. DA experiments were performed over 12 stations where daily continuous SWE measurements were acquired during 4 winters (2012-2016). Best SWE estimates are obtained with the assimilation of the $T_B$s at 11, 19 and 37 GHz in vertical polarizations. The overall SWE bias is reduced by 71% compared to original SWE simulations, from 23.7 kg m$^{-2}$ without assimilation to 6.9 kg m$^{-2}$ with the assimilation of the three frequencies. The overall SWE relative percentage of error (RPE) is 14.6% for sites with a fraction of forest cover below 75%, which is in the range of accuracy needed for hydrological applications. This research opens the way for global applications to improve SWE estimates over large and remote areas, even when vegetation contributions are up to 50% of the PMW signal.

30 **Keywords**: passive microwave, Crocus snowpack evolution model, DMRT-ML radiative transfer model, vegetation contributions, SWE retrievals, Eastern Canada, Assimilation scheme, Particle Filter



# 1 Introduction

In Québec, Eastern Canada, snowmelt runoff has become a major economic issue and plays a considerable role in flood events (Perry, 2000). Good forecasting of this water supply is essential to optimizing the management of hydroelectric dams. The amount of water stored in a snowpack is estimated by the snow water equivalent (SWE). Accurately predicting the evolution of the SWE is challenging over large and remote areas due to the high spatial and temporal variability of the snowpack and to the lack of *in situ* data, which are time-consuming and expensive to measure. Current operational hydrological forecasting models used by Hydro-Québec, one of the larger energy producers in North America, rely on surface snow surveys measurement interpolation (Tapsoba et al., 2005). It has been shown that the highest uncertainties in hydrological forecasting related to snow result from a lack of accurate estimates of the amount of snow accumulated during the winter season over large area (Turcotte et al., 2010). To have a better knowledge of the spatial distribution of the SWE, many approaches use snowpack models to simulate the evolution of the snow cover in response to meteorological conditions (Brun et al., 1989; Jordan, 1991; Lehning et al., 2002). Nevertheless, the use of models is challenging due to the imperfect knowledge of meteorological forcing data (Raleigh et al., 2015) (because of the low number of weather stations in remote areas) and simplifications of snow physics used in the models (Foster et al., 2005).

The assimilation of satellite observations is a promising approach used to reduce these uncertainties related to the lack of *in situ* data (Pietroniro and Leconte, 2005; Durand et al., 2009; Touré et al., 2011; De Lannoy et al., 2012; DeChant and Moradkhani, 2011, Kwon et al., 2017). In particular, passive microwave (PMW) satellite observations, which measure brightness temperatures ('$T_B$'), are sensitive to the volume of snow and provide information at a good temporal and spatial coverage (Hallikainen, 1984; Chang et al., 1996; Tedesco et al., 2004). It has been shown that the assimilation of PMW satellite data into snow models added valuable information in order to compensate for initialization errors and to improve SWE simulated by snow model (Sun et al., 2004). These approaches appear to be very promising to evaluate and predict water resources but are still under development to be further used for operational hydrological applications (Xu et al., 2014). Larue et al. (2017) has shown that the GlobSnow-2 SWE product (Takala et al., 2011), which assimilates both $T_B$ satellite data and local snow depth observations, was not accurate enough for hydrological modeling, mainly because of its dependence on *in situ* data in remote areas.

The main difficulty in the assimilation of PMW satellite observations in boreal forest areas is to quantify all the contributions that affect the measured signal. PMW satellite observations have a low spatial resolution (~ 10 x 10 km$^2$) and many contributions are measured by satellite sensors, in addition to the PMW emission from the volume of the snowpack (vegetation canopy, ice crust, frozen/unfrozen soil, lakes, moisture in the snow, topography, etc.) (Kelly et al., 2003; Koenig & Forster, 2004). In boreal areas, the PMW emission from the forest canopy within a pixel can contribute up to half of the PMW signal measured by satellite sensors (Roy et al., 2012, 2016). This contribution does not only depend on the fraction of forest cover, but also on the biomass (liquid water content), the vegetation volume and the structure of the canopy (stem, leaf, trunk) (Franklin, 1987). To adjust snowpack model simulations, several studies suggest using radiative transfer models, coupled to a



snowpack model, to take into account the different contributions to the PMW signal at the top of the atmosphere and to directly assimilate PMW satellite observations (Brucker et al., 2011; Durand et al., 2011; Langlois et al., 2012; Roy et al., 2016).

This paper aims at developing and validating the assimilation of PMW satellite observations for SWE improvements over Québec by adjusting meteorological forcing data and simulated snowpack states without using any surface-based observations.

AMSR-2 satellite sensors provide the $T_B$ observations at 11, 19 and 37 GHz. The data assimilation scheme (DA) is a Sequential Importance Resampling Particle filter (referred to as PF-SIR). The PMW emission from the snowpack is computed by using the Crocus snowpack model (Brun et al., 1989) coupled to a microwave snow emission model, the Dense Media Radiative Transfer - Multi Layers model (DMRT-ML) (Picard et al., 2013). This scheme is further referred as the Crocus/DMRT-ML chain and has been previously calibrated over Québec (Larue et al., 2018). This implementation was first validated in Larue et

al. (2018) by using synthetic observations. For the assimilation of satellite data, the challenge is to accurately simulate the $T_B$ measured at the top of the atmosphere ($T_{B\ TOA}$) by including contributions other than snow (soil, vegetation and atmosphere). The vegetation transmissivity model ($\omega$-$\tau_{opt}$), the soil emission model of Wegmüller and Mätzler (1999, WM99) and the atmospheric emission model of Liebe (1989) are added and calibrated to simulate the PMW emission of satellite observations (Roy et al., 2015).

The specific objectives of this paper are thus to: 1) calibrate the soil and the vegetation radiative transfer models coupled with the Crocus/DMRT-ML chain to simulate $T_{B\ TOA}$ over several years (2012 to 2016); and 2) evaluate the performance of the assimilation of PMW data in Crocus using SWE measurements obtained over twelve reference nivometric stations from 2012 to 2016. This paper opens the way to a functional spatialized method for improving SWE estimates over large and remote areas without using surface-based data.

**2 General framework**

### 2.1 Study area and evaluation database

Figure 1 shows the region of interest located in the province of Québec, Eastern Canada area (46-56°N). This area includes the watershed of La Grande, in the middle north of Québec (below 56°N), the watersheds of the Outaouais and of the Mauricie in the central area of Québec (46-48°N, see Fig. 1), which are equipped with SWE and snow depth sensors for hydrological

purposes. Québec is characterized by different eco-climatic conditions, mainly constituted of forested area (dense boreal forests with coniferous and deciduous), and a flat topography.





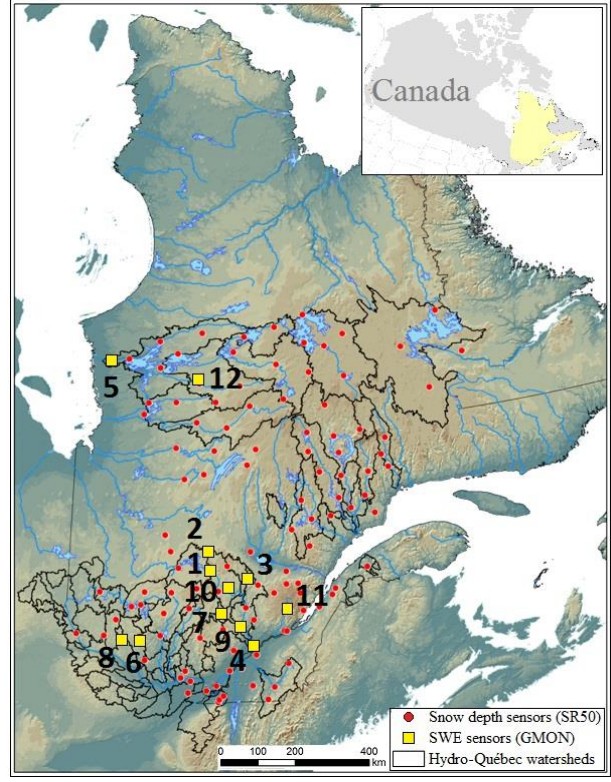

**Figure 1. SWE measurement stations with the 'GMON' SWE sensors (yellow squares, see Table 1) in the province of Québec, Eastern Canada. The red circles are the snow depth sensors ('SR50') used by Hydro-Québec for hydrological purposes, overlaid on a relief map (from blue-low to brown-higher altitudes) and watershed contours (black lines). The LG watershed is located in the middle north of Québec, the Outaouais and the Mauricie watersheds are in southwestern and south-central Québec, respectively.**

To evaluate SWE simulations, SWE measurements were taken from 2012 to 2016 by twelve nivometric stations (see numbered stations on Fig. 1), located through a north-south gradient in Québec. This database is fully described in Larue et al. (2018). Table 1 describes the main station characteristics including the mean maximum SWE measured values over operating periods. Daily SWE measurements are derived from gamma ray SWE sensors (Campbell Scientific CS725, 'GMON') with an average error of +5% (Choquette et al., 2008). Two stations (Nos. 5 and 12) are located in the subarctic eco-climatic zone (53-54°N, James Bay area), eight in the coniferous boreal zone (46-48°N) and two (Nos. 4 and 11) in a mixed forest area in southern Québec (45.3°N). Sensors were calibrated by Hydro-Québec from numerous field measurement campaigns during the first year following their installations.





**Table 1. Characteristics of the nivometric stations: SWE (in kg m$^{-2}$) data, Latitude (Lat.), Longitude (Long.) and Elevation (El., a.s.l. in meters) of stations, Dist. GEM-station is the distance between the station and the center of the associated GEM grid cell, time period of observations, average of the maximum observed data over the studied period, data providers (HQ: Hydro-Québec, U. Sherb: Université de Sherbrooke, U. Laval: Université Laval).**

| Sites # | Lat. | Long. | El. | Dist. GEM-station (km) | Time period | Mean maximum SWE value (kg m$^{-2}$) | Data provider |
|---|---|---|---|---|---|---|---|
| 1 | 48.3 | -74.1 | 100 | 3.4 | 2012-2016 | 272 | HQ |
| 2 | 48.9 | -74.2 | 100 | 4.9 | 2012-2016 | 277 | HQ |
| 3 | 47.9 | -72.9 | 100 | 4.7 | 2012-2016 | 252 | HQ |
| 4 | 46.6 | -72.8 | 136 | 4.2 | 2012-2016 | 253 | HQ |
| 5 | 53.7 | -78.2 | 103 | 4.2 | 2012-2016 | 213 | HQ |
| 6 | 46.7 | -76.0 | 229 | 2.3 | 2012-2016 | 161 | HQ |
| 7 | 47.0 | -74.3 | 469 | 3.3 | 2012-2016 | 235 | HQ |
| 8 | 46.9 | -76.4 | 330 | 1.8 | 2012-2016 | 212 | HQ |
| 9 | 46.9 | -73.7 | 372 | 1.9 | 2012-2016 | 180 | HQ |
| 10 | 47.7 | -73.6 | 398 | 3.5 | 2012-2016 | 202 | HQ |
| 11 | 47.3 | -71.2 | 669 | 2.6 | 2015-2016 | 396 | U. Laval |
| 12 | 53.4 | -75.0 | 389 | 4.0 | 2014-2016 | 211 | U. Sherb |
| Mean | | | | 3.4 | 2012-2016 | 237 | |

## 2.2 General setup

Figure 2 shows the general methodology developed to simulate and to assimilate AMSR-2 satellite observations into the snowpack model.

To simulate the signal measured by satellite sensors at the top of the atmosphere ($T_{B\ TOA}$), a chain of models was implemented and calibrated over Eastern Canada. The three hourly-continuous atmospheric forcing database provided by the Global Environmental Multiscale weather prediction model (referred to as 'GEM'; Coté et al., 1998) was used to drive the multi-layer Crocus snowpack model (described in Sect. 3.2.1 further). Each GEM grid cell has a spatial resolution of 10 x 10 km$^2$, which is on the same order as the observation scale. In this study, the Crocus model computes the evolution of the snowpack (SWE, snow depth, density, etc.) each day at 1 pm, in agreement with the AMSR-2 observation time (Sect. 3.1.1). The DMRT-ML radiative transfer model (Sect. 3.2.1), driven with Crocus outputs, was used to simulate the PMW emission from the modeled snowpack (referred to as '$T_{Bsnow}$') at 11, 19 and 37 GHz, at vertical and horizontal polarizations ('V-pol' and 'H-pol', respectively). The contribution of the atmosphere was estimated by using an atmospheric model (Liebe, 1989) driven with the total of precipitable water integrated over 28 atmospheric layers and provided by GEM (Dolant et al., 2016) (Sect. 3.3). The surface emissivity for a rough soil was deduced by calibrating the soil model of Wegmüller and Mätzler (1999, WM99) and





the contributions of the vegetation were quantified with the (ω-$\tau_{opt}$) radiative transfer model (Sect. 3.3). To take into account the variability of the canopy emissivity, the calibration of the (ω, $\tau_{opt}$) parameters were linked to the 4-day leaf area index (LAI) product from MODIS data (1 x 1 km$^2$), averaged for each AMSR-2 grid cell (10 x 10 km$^2$) (Sect. 3.3.3). These calibrations of soil and vegetation parameters were performed over the summer period to avoid the bias due to the presence of

the snowpack.

The brightness temperatures ($T_{Bs}$) measured by AMSR-2 satellite sensors were assimilated in a data assimilation (DA) scheme (see Sect. 3.4). Raleigh et al. (2015) have shown that meteorological forcing data were the major sources of errors in snow model simulations. Hence, we assume here that the uncertainties of GEM meteorological forcing data are the only sources of errors in the $T_B$ modeling. Quantifying the modeling errors due to physical simplifications inside the model is very difficult

due to the observation spatial scale. Further studies are needed to estimate these errors over the study area and to take it into account in the DA experiment. The observations errors were assumed to be known and the modeling errors were estimated by perturbing selected meteorological forcing variables. An ensemble of 150 $T_B$ simulations was obtained and the distribution of these 'prior estimates' represent the modeling error in response to GEM uncertainties. . A Particle filter with an SIR algorithm was used in the DA scheme to update the simulated $T_{B\ TOA}$ over the winter by adjusting meteorological forcing data and

snowpack states (posterior estimates) when an observation was available (Fig. 2). Several configurations of the DA scheme were tested over three evaluation sites representing different environmental conditions and the best configuration of the DA scheme was evaluated over the 12 validation reference sites from 2012 to 2016 (Sect. 3.4).

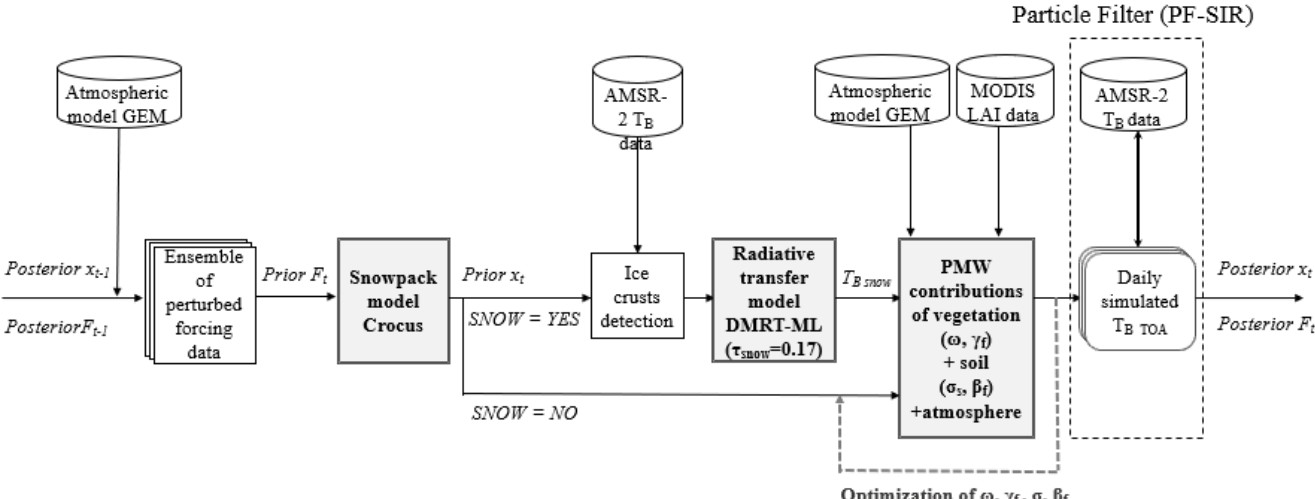

**Figure 2. Methodological scheme describing the DA scheme in the chain of models for SWE retrievals by updating perturbed atmospheric forcing data and snowpack states ('$F_t$' and '$x_t$', respectively, see Sect. 3.4).**



# 3 Materials and methods

## 3.1 Database

### 3.1.1 AMSR-2 observations

AMSR-2 satellite sensors (Imaoka et al., 2010) provide PMW satellite observations on the 11 (10.7), 19 and 37 GHz channels
at V-pol and H-pol. Images produced by AMSR-2 are freely available on the Japan Aerospace Exploration Agency (JAXA)
website. This study used the Level 3 Version 2 product, which provides daily $T_{Bs}$ normalized on a North Hemisphere polar
stereographic projection with a spatial resolution of 10 x 10 km$^2$ (see http://gcom-w1.jaxa.jp for the specifications of the
projection), from 1 August 2012 to 1 July 2016. $T_{Bs}$ from AMSR-2 are computed twice a day: at 1 pm (ascending pass) and at
1 am (descending pass). Only the ascending pass was used in this study (about 13:00 local time) since the snowpack was
computed once a day at 1 pm (local time). The use of the ascending pass allowed avoiding the nighttime refreeze process. To
reduce observation errors due to the daytime melting process, the approach was evaluated during the dry snow period, from
December to mid-March. This aspect is further discussed in Sect. 5.1.

### 3.1.2 LAI MODIS data

The 4-day LAI product provided by MODIS TERRA data (MOD15A3; Myneni et al., 2002) was used to characterize the
vegetation contributions on the total emissivity (Fig. 2). The product has a spatial resolution of 1 x 1 km$^2$ and was resampled
on the AMSR-2 grid of 10 x 10 km$^2$ by averaging all LAI data within each AMSR-2 grid cell (referred to as 'LAI$_{AMSR-2}$'). For
each site, Table 2 describes the summer and winter average values ('LAI$_{summer}$' and 'LAI$_{winter}$') calculated using LAI$_{AMSR-2}$
from 1 July to 31 August and from 1 January to 1 March over the 2012 to 2016 time period, respectively (Roy et al., 2014).





**Table 2. LAI$_{summer}$ is the mean of the LAI provided by MODIS for the July-August time period and averaged over the AMSR-2 grid cell (10 x 10 km$^2$), LAI$_{winter}$ is the mean LAI for the January-March time period. $f_{cover}$ is the fraction of forest cover within the AMSR-2 grid cell extracted from the land cover map Circa 2000 (Sect. 3.3). The percentages of coniferous, deciduous and water areas are the percentages distributed within the $f_{cover}$. Sites are ranked in the increasing order of $f_{cover}$. The three highlighted sites (gray cells) are the sites selected to test the configuration of the DA scheme in Sect. 4.2.**

| Site # | LAI$_{summer}$ | LAI$_{winter}$ | $f_{cover}$ (%) | Coniferous (%) | Deciduous (%) | Water (%) |
|---|---|---|---|---|---|---|
| #12 | 1.07 | 0.04 | 24.2 | 77.6 | 14.4 | 4.9 |
| #5 | 1.07 | 0.08 | 31.5 | 66.5 | 25.9 | 7.0 |
| #4 | 2.63 | 0.06 | 47.6 | 8.5 | 70.3 | 1.4 |
| #7 | 3.13 | 0.28 | 59.3 | 49.9 | 45.8 | 4.0 |
| #10 | 2.47 | 0.17 | 61.8 | 67.3 | 30.1 | 2.4 |
| #1 | 2.96 | 0.28 | 63.7 | 41.6 | 55.8 | 2.2 |
| #3 | 3.69 | 0.25 | 65.5 | 44.6 | 52.1 | 3.3 |
| #2 | 1.99 | 0.12 | 66.6 | 79.4 | 16.6 | 3.5 |
| #8 | 4.11 | 0.22 | 72.1 | 15.5 | 80.2 | 4.3 |
| #11 | 2.43 | 0.19 | 74.5 | 52.5 | 46.6 | 0.5 |
| #6 | 2.82 | 0.11 | 81.5 | 18.1 | 75.3 | 6.5 |
| #9 | 3.65 | 0.43 | 84.0 | 60.9 | 36.1 | 2.9 |

### 3.1.3 Land cover map of Canada

The land cover map of Canada Circa 2000 (available at http://www.geobase.ca/geobase/en/data/landcover/index.html) (referred to as 'LCC') was used to extract the fraction of forest cover ('$f_{cover}$') within each AMSR-2 grid cell. This product

10    provides the percentage of coniferous, herbaceous, deciduous and water areas with a spatial resolution of 1 x 1 km$^2$ and was resampled to generate average values within each 10 x 10 km$^2$ AMSR-2 grid cell. Table 2 shows the fractions of forest cover provided by the LCC and resampled over AMSR-2 grid cells for each site. As expected, Sites 5 and 12, which are located in the subarctic area (Fig. 1), have a low $f_{cover}$ (below 32%). The other sites in boreal areas have an $f_{cover}$ of up to 60%. Sites 6 and 9 are in particularly dense forest areas, with a high $f_{cover}$ (up to 80%). In such dense forest areas, the signature of the

15    underlying snow can be significantly attenuated during the winter period and bias the measured T$_B$ signal. To test the configuration of the DA scheme for several environmental conditions, the T$_B$ assimilations for Site 12 ($f_{cover}$ = 24.2%), Site 1 ($f_{cover}$= 63.7%) and Site 9 ($f_{cover}$ = 84.0%) were analyzed in a preliminary experiment (Sect. 4.2.1).





Moreover, the presence of lakes can affect the PMW signal. Lake ice (when snow cover is absent) increases the PMW signal at high frequencies and, at low frequencies, the contribution of water bodies acts as a reflector and the emissivity remains low (De Sève et al., 1999). With snow cover on lakes, the different snow states on the lakes compared to snow cover under forest also modified the emitted signal (see Derksen et al. 2012, 2014). Nevertheless, we made the hypothesis that these impacts were negligible over our studied sites, which have lake water fractions under 7% within their AMSR-2 grid cells (Table 2) (masks are generally applied for water fractions of up to 20%, Takala et al., 2011).

## 3.2 Simulation of the PMW emission from the snowpack

### 3.2.1 Coupling of Crocus and DMRT-ML

The chain of models developed to simulate $T_{Bsnow}$ is identical to that of Larue et al. (2018), so only a brief description of the approach is detailed here (see Fig. 2).

To generate a three hourly-continuous meteorological forcing database for running Crocus, successive GEM forecasts were taken from the +09 forecast hour to the +18 forecast hour provided at the 00 and 12 UTC analysis time of each day. The Crocus snowpack evolution model (Brun et al., 1989, 1992; Vionnet et al., 2012) is coupled with the ISBA land surface model within the SURFEX interface (Surface Externalisée, in French) (Decharme et al., 2011; Masson., 2013). SURFEX/ISBA/Crocus (hereafter referred to as "Crocus") computes the evolution of the physical properties of the snowpack and the underlying ground (soil). In particular, Crocus represents the detailed snow microstructure evolution in time through the formulations in Carmagnola et al. (2014). The number of snow layers is dynamic and evolved according to physical properties updated at each time step. The maximum number of simulated snow layers was fixed at 15 in this study, as a compromise between accuracy and computing time (not shown). Configuration and initialization of the Crocus snowpack model are the same as described in Larue et al. (2018).

$T_{B\,snow}$ was computed by driving the radiative transfer model DMRT-ML with Crocus outputs. The DMRT-ML model is well-detailed in the literature (Tsang et al., 1992; Tsang and Kong, 2001; Picard et al., 2013), so only the calibration is described here. Snow grain size, and more generally snow microstructure, are factors that most affect the accuracy of simulated PMW emission from a snowpack as they determine the strength of scattering mechanisms in the snowpack at the high frequencies used (Roy et al., 2013; Leppänen et al., 2015; Sandells et al., 2017, Larue et al., 2018). In DMRT-ML, snow grains are represented as spheres of ice with variable interactions between them. The potential formation of clusters of grains, which increases the effective snow grain size, is not taken into account, generating uncertainties (Picard et al., 2013). Several studies have shown that DMRT-ML needed an effective scaling factor to represent the stickiness between snow grains and to correct the snow microstructure representation (Brucker et al., 2011; Roy et al., 2013; Royer et al., 2017). Larue et al. (2018) have shown that a mean snow stickiness parameter ($\tau_{snow}$) of 0.17 was optimal to simulate $T_{Bsnow}$ over boreal snow in Québec (RMSE of 27 K) when DMRT-ML is driven by Crocus snow profiles. This constant $\tau_{snow}$ value was used in the implemented chain of





models. Nevertheless, this effective parameter could change with snow type (Royer et al., 2017; Larue et al., 2018). The use of the $\tau_{snow}$ parameter as a free variable in the DA scheme is discussed in Sect. 5.2.

### 3.1.2 Ice lens detection algorithm

Since ice lenses ('IL') within a snowpack significantly reduce $T_B$ mainly at H-pol (Montpetit et al., 2013; Roy et al., 2016),
ice layers must be detected and added in the simulated Crocus snow profiles to improve $T_{Bsnow}$ simulations. $T_B$ in H-pol are much more attenuated by the presence of an IL than $T_B$ in V-pol, since the coefficient of reflectivity is stronger in H-pol (Montpetit et al., 2013). Therefore, by following the daily evolution of the PMW emission from the snowpack with AMSR-2 observations, the formation of an IL can be detected by using a threshold on the polarization ratio PR defined by Cavalieri et al. (1984) for a given frequency ($\nu$),

$$PR(\nu) = \frac{T_B(\nu, V-pol) - T_B(\nu, H-pol)}{T_B(\nu, V-pol) + T_B(\nu, H-pol)} \qquad (1).$$

In this study, an IL was inserted on the top of the simulated snowpack if the AMSR-2 PR(11) was above 0.06 (Roy, 2014). To integrate the IL in the snow profile, a 1-cm layer with a density of 900 kg m$^{-3}$ and snow grain radius set to zero was first added at the surface of the snowpack when it was detected (Roy et al., 2016). The difficulty knowing how to evolve this IL in the snowpack. The Crocus snowpack model has not yet been adapted to integrate the formation of ILs and evolve them in a
coherent way (Quéno *et al.*, 2016). Nevertheless, it was shown in Larue et al. (2018) (from field measurements) that an IL of 1 cm located 4 cm from the surface in the simulated Crocus snow profile minimized the bias of DMRT-ML simulations due to the presence of an IL in the snowpack (regardless of its real location in the snow profile). Hence, as soon as a snowfall is detected with GEM precipitation data, the IL firstly added on the top of the surface was positioned 4 cm from the surface in the simulated snow profile. The maximum number of detected IL was fixed at two. In this case, the first detected IL was
positioned at 8 cm from the surface and the second at 4 cm after a snowfall was detected. During winter 2014-2015, one IL was detected at sites 1 and 12 (22 December 2014 and 15 December 2014). At site 9, two ILs were detected: one on 10 December 2014 and another on 1 January 2015.

### 3.3 Simulation of the PMW emission at the top of the atmosphere

The PMW brightness temperature ($T_B$) emitted by a AMSR-2 grid cell can be written as (2),

$$T_B = \left[ f_{season} \cdot T_{B\,forest} + (1 - f_{season}) T_{B\,open} \right] \qquad (2)$$

where $f_{season}$ is the seasonal (winter or summer) fraction of forest cover in the AMSR-2 grid cell, $T_{B\,forest}$ is the PMW emission with vegetation contributions and $T_{B\,open}$ is the PMW emission without vegetation contributions. The $f_{cover}$ values provided by the Circa 2000 map are constants whereas these fractions of forest evolve according to the season. To take into account the temporal evolution of the forest cover for the winter and summer periods (defined as the time period with and without snow,
respectively) and to estimate the $f_{season}$ used in Eq. (2), $f_{cover}$ was linked respectively to LAI$_{winter}$ and to LAI$_{summer}$ by comparing



the $f_{cover}$ map to the two resampled maps (both resampled on the AMSR-2 projection) throughout Québec area (not shown). The seasonal fraction of $f_{cover}$ are related to seasonal LAIs with the Eq. (3) and (4) for summer and winter respectively,

$$f_{summer} = 0.9 * (1 - \exp(-2.7 * LAI_{summer}))^{3.2} \tag{3}$$

$$f_{winter} = 0.9 * (1 - \exp(-16.0 * LAI_{winter}))^{0.3} \tag{4}$$

The linear correlation between the $f_{summer}$ values estimated from the LCC and the $f_{summer}$ values fitted to LAI data with the Eq. (3) has a coefficient correlation r equal to 0.94 and a p-value below 0.01. For the LCC $f_{winter}$ values and the $f_{winter}$ values fitted to the LAI data (see Eq. (4)), the coefficient correlation r is equal to 0.95 and the p-value is below 0.01.

### 3.3.1 Vegetation contributions

The PMW emission from the vegetation varies according to the forest characteristics, such as the biomass, the structure of the
vegetation or the liquid water content of the canopy. In this study, the vegetation contribution was modeled according to the simplified radiative transfer model ($\omega$-$\tau_{opt}$) (Mo et al., 1982), where the parameters are estimated by fitting the simulated $T_{Bs}$ with observations (Grant et al., 2008, Roy et al., 2012). The $\omega$ is the simple scattering factor of the albedo. Given the incidence angle $\theta = 55°$ of AMSR-2 satellite sensors, the optical thickness of the vegetation $\tau_{opt}$ is a function of the forest transmissivity ($\gamma$) such that $\gamma = \exp(-\tau_{opt} / \cos\theta)$. The forest transmissivity, which varies according to the frequency ($\nu$) used and is further
called $\gamma_v$. At the satellite sensor, the expression of $T_{B\ TOA}$ in boreal areas was described by the Eq. (2), which can be detailed with the Eq. (5) and (6) (see Roy et al., 2012),

$$T_{B\ forest} = \left[ \gamma_v . e_{surf} . T_{surf} + (1 - \omega) . (1 - \gamma_f) . T_{veg} + \gamma_v . (1 - e_{surf}) . (1 - \omega) . (1 - \gamma_v) . T_{veg} + (1 -$$

$$e_{surf}) . \gamma_v^{\ 2} . T_{B\ atm\downarrow} + (1 - \gamma_v) . \omega . T_{B\ atm\downarrow} \right] . \gamma_{atm} + T_{B\ atm\uparrow} \tag{5}$$

$$T_{B\ open} = \left[ e_{surf} . T_{surf} + (1 - e_{surf}) . T_{B\ atm\downarrow} \right] . \gamma_{atm} + T_{B\ atm\uparrow} \tag{6}$$

where $T_{surf}$ is the surface temperature, $e_{surf}$ is the surface emissivity under the canopy (with or without snow) for a given frequency, $T_{veg}$ is the temperature of the vegetation (taken as equal to the air temperature at 2 meters, provided by GEM). $T_{Batm\downarrow}$ and $T_{Batm\uparrow}$ are respectively the descending and ascending atmospheric contributions, and $\gamma_{atm}$ is the transmittance of the atmosphere. These atmospheric contributions were modeled using the Liebe (1989) model implemented in the Helsinki University of Technology (HUT) snow emission model (Pulliainen et al., 1999). The model considers radiative transfer through
the atmospheric layers and provides values $T_{Batm\downarrow}$, $\gamma_{atm}$ and $T_{Batm\uparrow}$ at the satellite sensor level (Liebe, 1989) according to the precipitable water integrated for all atmospheric layers provided by GEM (Dolant et al., 2016). Thus, for snow free conditions, only forest ($\omega$, $\gamma_v$) and soil ($e_{surf}$) parameters are unknown and need to be adjusted for each site by fitting the outputs of the model according to the observations.



### 3.3.2 Soil contributions

To deduce the surface emissivity for a rough soil ($e_{surf,p}$ for a given polarization p), the soil model of Wegmüller and Mätzler (1999, WM99) was used to calculate the surface reflectivity for a rough soil throughout the year under the canopy ($r_{surf,p}$ for a given polarization p), with or without snow by using the Eq. (7) and (8),

$$r_{surf,H} = 1 - e_{surf,H} = \Gamma_{Fresnel,H} . \exp(-\sigma_s^{\sqrt{0.1.cos\theta}}) \tag{7}$$

$$r_{surf,V} = 1 - e_{surf,V} = r_H . \cos\theta^{\beta} \tag{8}$$

$r_{surf,p}$ mainly depends on the surface roughness and Fresnel coefficients ($\Gamma_{Fresnel, H}$). In the Eq. (7), the simplified parameter $\sigma_s = k.\sigma$ was used, where k is the wave number and $\sigma$ the standard deviation of the surface height (in meters). When the soil is frozen, parameters derived from Montpetit et al. (2017) were used (see Sect. 4.1). When the soil is not frozen, $\Gamma_{Fresnel,H}$ was estimated from the dielectric constant calculated with the Dobson (1985) equations according to the soil moisture and the soil temperature. These variables are daily computed with the Crocus model, coupled to the ISBA land surface model. The soil reflectivity in vertical polarization also depends on a parameter $\beta$ (Montpetit, 2015), which describes the polarization of the signal and varies according to the frequency used (referred to as $\beta_v$ hereafter). Hence, the soil parameter $e_{surf}$ is linked to the couple ($\sigma_s$, $\beta_v$) and mainly evolved according to soil moisture and soil temperature.

### 3.3.3 Inversions of vegetation and soil parameters

The inversion of forest ($\omega$, $\gamma_v$) and soil ($\sigma_s$, $\beta_v$) parameters was carried out in summer to avoid the bias due to the presence of a snowpack. Forest parameters ($\omega$, $\gamma_v$) depend on the forest characteristics, such as the biomass and the structure of the canopy for each site. To take into account the temporal variations of these caracteristics, the forest parameters were linked to the LAI. It also allowed a realistic continuity of the ($\omega$, $\gamma_v$) calibration for the winter period. Using the vegetation water content equation defined by Pampaloni and Paloscia (1986), the parameter $\gamma_v$ is related to the 4-day LAI for a given frequency $v$ with the relation (9),

$$\gamma_v = e^{-b.k^a.\left(\exp\left(-\frac{LAI}{3}\right)-1\right)/cos\theta} \tag{9}$$

where a and b are two constants to calibrate. To reduce the number of unknown variables, the Eq. (9) has been simplified to use only one constant $\eta_v$ such as $\eta_v = e^{-b.k^a}$.

The vegetation and soil parameters were inverted by minimizing the difference between simulated $T_{B\ TOA}$ compared to $T_{Bs}$ measured with AMSR-2 sensors at 11, 19 and 37 GHz in vertical polarizations. The same approach was developed in Roy et al. (2014) and adapted for PMW emission in boreal areas. It has been shown that the soil and vegetation contributions are strongly linked and can not be decoupled. Moreover, the two parameters ($\omega$, $\sigma_s$) have been shown to be constant in frequency in previous studies. Pellarin et al. (2006) have shown a $\omega = 0.06$ for coniferous forest at 6.6 and 11 GHz and Meissner and Wentz (2010) have shown that the increase of $\omega$ from 6.6 to 19 GHz was weak. Therefore, for each site and for each ($\omega$, $\sigma_s$) value couple (considered constant in frequency), optimal ($\eta_v$, $\beta_v$) values were optimized for each frequency (at 11, 19 and 37 GHz) in V-pol. Optimizations of ($\eta_v$, $\beta_v$) parameters have been tested in H-pol and V-pol and parameters were assumed to be



constants in polarizations (not shown). These ($\eta_v$, $\beta_v$) optimizations were carried out for each couple ($\omega$, $\sigma_s$) values, which varied iteratively with a step of 0.01 (from 0.02 to 0.16) and 0.05 (from 0.01 to 1.1), respectively (Roy et al., 2014).

## 3.4 Data assimilation setup

As a first step, the previous study of Larue et al. (2018) tested the feasibility of the DA scheme in a controlled environment by using synthetic $T_{Bsnow}$ observations, obtained by running the Crocus/DMRT-ML chain (Fig. 2) with one perturbed meteorological forcing data. The results showed an SWE ensemble RMSE reduced by 82% with the multi-variate assimilation of differences between $T_{Bs}$ at 19-37 GHz and 19-11 GHz in vertical polarizations, compared to SWE ensemble RMSE without assimilation ('open loop runs'). In the present study, the same DA setup as described in Larue et al. (2018) was implemented except that real satellite observations were used. The observation errors are difficult to quantify due to the difference between model and observation representativeness. A poor parameterization of observation error statistics quickly leads to ensemble degeneracy, i.e. an ensemble collapsing to a unique particle. To avoid ensemble degeneracy, an inflation technique of the covariance matrix of observation errors (R matrix) is developed and implemented.

### 3.4.1 DA framework

The DA scheme is a particle filter with a Sequential Importance Resampling algorithm (hereafter referred to as 'PF-SIR') that is well-documented in Van Leeuwen (2009, 2014) and Gordon et al. (1993) and relatively easy to implement with a snowpack model (Dechant & Moradkhani, 2011; De Lannoy et al., 2012; Charrois et al., 2016; Larue et al., 2018). The PF-SIR represents the probability density function (pdf) of the model state with an ensemble of states (called particles), which is updated when an observation is available. An ensemble approach is preferred because of the non-linearity of the system. Moreover, the particle filter approach can cope with the variable number of state variables resulting from the changing number of snow layers in Crocus. The created ensemble represents uncertainty in SWE and in $T_B$ simulations due to the uncertainties of meteorological inputs (Fig. 2).

The daily ensemble of meteorological forcing data was created by perturbing selected GEM data (air temperature, wind speed, precipitation and short and long wave radiations) according to their respective uncertainties estimated in Larue et al. (2018). Meteorological forcing perturbations are evolved in time following a first-order autoregressive process to simulate their realistic temporal variations (Charrois et al., 2016). Precipitation, wind speed and short-wave radiations ('$SW_{down}$') were perturbed by a multiplicative factor centered at 1. Perturbation boundaries were fixed at -0.9 and 0.9. The air temperature was perturbed by an additive factor, with boundaries fixed at -3 K and +3 K. Perturbed long wave radiations ('$LW_{down}$') were estimated according to perturbed $T_{air}$ from a linear regression estimated in Larue et al. (2018). In order to maintain physical consistency in the simulations, $SW_{down}$ was limited to 200 $W.m^{-2}$ when there was precipitation (presence of clouds) (Charrois et al., 2016). The ensemble was composed of 150 members, which was found to be adequate in Larue et al. (2018). Note that this ensemble method is stochastic and was chosen to be easily implemented and tested for each site, which represent



independent grid-cells. Further studies would be necessary to validate the spatialization of this approach by initializing the chain of models with coherent ensemble forecasts.

The snowpack prior state $x_t$ at time t is computed according to the updated past state of snowpack simulations at time t-1 (posterior state $x_{t-1}$) and to the prior perturbed meteorological forcing data $F_t$ from time t-1 to t (see Fig. 2). The predicted observation is computed with

$$y_t{}^i = h(x_t{}^i) \qquad\qquad (10)$$

where $y_t{}^i$ is $T_{B\,TOA}$ predicted from particle i (i=0..N, with N the ensemble size). The observation operator $h$ is the $\tau_{snow}$-calibrated DMRT-ML model and the calibrated radiative transfer models estimating soil, atmosphere and vegetation contributions. In the analysis step, the new posterior distribution is updated by weighting each particle $x_t{}^i$ according to the distance between $y_t{}^i$ and the AMSR-2 $T_B$ observation. With the SIR algorithm, the pdf is resampled by duplicating particles with large weights (i.e., close to observations) and taking off those with negligible weights (far from observations). With the Arakawa procedure used here for ensemble resampling (Arakawa, 1996; same as Charrois et al, 2016), a particle is definitely selected if its weight is larger or equal to the inverse of the ensemble size (N=150). The observation error standard deviation associated with AMSR-2 observations was assumed to be 2 K (Durand & Margulis, 2006, 2007).

Ensemble resampling considerably reduces the risk of degeneracy, but does not eliminate it. Degeneracy starts when only a few particles have significant weights. These particles are selected many times, leading to a loss of diversity of the posterior ensemble. After several assimilation steps, the ensemble quickly reduces to a single particle. Ensemble degeneracy can be detected when the number of selected particles (those with high weights) is below an effective limit number $N_{keep}$, here fixed at 25 as a compromise between the quality of the DA scheme and the size of the ensemble (not shown). Hence, to avoid a degeneracy problem, the weight of the 25-th selected particle ($we_{keep}$) must always be larger or equal to the inverse of the ensemble size (N=150). In this study, we developed a new technique to ensure this, which consists in the online adjustment of the observation error covariance matrix such that $we_{keep}$ is at least equal to 1/N. The rationale here is that, because the weights are nonlinear functions of the observation error covariance matrix, a larger matrix tends to flatten the distribution of weights and favours the selection of more particles. This adjustment is performed with an inflation of the initial matrix, and the detailed algorithm is provided in Appendix A. Ensemble degeneracy is often caused by extreme precipitation events resulting in very high $T_B$ values difficult to represent with the model. The online adjustment technique mitigates the consequences of this model deficiency on the snow simulations over the rest of the season. The other side of the coin is that a "good" observation can be ruled out if the model is not able to reproduce it, thereby reducing the accuracy of the snowpack estimation.

### 3.4.2 Experimental setup

In a first step, three DA experiments were tested over sites 1, 9 and 12 for winter 2014-2015 (Sect. 3.1.3) to analyze the sensitivity of the DA scheme for SWE improvements according to the assimilated frequencies: a) assimilation of the $T_B$ differences between 19 and 37 GHz and between 19 and 11 GHz, in V-pol (referred to as '$\Delta T_{B,19\text{-}37}$' and '$\Delta T_{B,19\text{-}11}$', respectively); b) assimilation of $\Delta T_{B19\text{-}37}$ only; c) assimilation of the three $T_B$s at 11, 19 and 37 GHz in V-pol. While the DA of



$T_{Bs}$ at 11, 19 and 37 GHz in V-pol should give the best results since this combination of frequencies imposes more constraints, the risk of encountering a degeneracy problem is higher. The combination of both $\Delta T_{B,19-37}$ and $\Delta T_{B,19-11}$ is commonly used in the literature for SWE retrievals (Chang et al., 1987; Tedesco et al., 2004; Tedesco & Nervekar, 2010). The assimilation of the $\Delta T_{B,19-37}$ only was also studied to analyze the sensitivity of $T_B$ assimilation for deep snowpack when $T_{B,37}$ saturates for a SWE

up to about 150 mm (Mätzler et al., 1994) and to evaluate the supply of information from 11 GHz in the assimilation of both $\Delta T_{B,19-37}$ and $\Delta T_{B,11-19}$ for SWE improvements. Here we used V-pol $T_B$ because H-pol $T_B$ is more sensitive to the stratigraphy of the snowpack and to the presence of ILs (Mätzler, 1987).

DA experiments were applied between 1 November and 1 May. To avoid wet snow conditions, the DA is not performed when a liquid water content is observed in the modeled snowpack. This variable is estimated from Crocus, driven with original

meteorological forcing data. SWE were evaluated over both the dry snow period (from 1 December to 15 March) and the whole winter (when a snowpack is detected).

To quantify the performance of the DA scheme, the daily RMSEs of ensembles of simulated SWE obtained with and without the DA scheme were compared by using the Eq. (11),

$$RMSE_t = \sqrt{\left(\frac{1}{N}\sum_{i=1}^{N}\left(X_{sim\,i,t} - X_{Obs\,t}\right)^2\right)} \qquad (11)$$

where N is the ensemble size, $X_{sim\,i,t}$ is the simulated variable from the member i at time t, and $X_{Obs\,t}$ is the diagnostic variable at time t obtained from AMSR-2 observations.

The best configuration of the DA scheme was then applied over the 12 sites, from 2012 to 2016. To estimate the accuracy for hydrological applications, the median of the SWE ensemble obtained with DA ($SWE_{DA}$) was compared to SWE measurements. The median was used instead of the mean to reduce the potential impact of extreme perturbations. The evaluation of the DA

scheme is performed by comparing $SWE_{DA}$ RMSE and the relative percentage of error ('RPE') values to the original SWE simulations ($SWE_{Crocus}$), obtained by driving Crocus with original meteorological forcing data. The relative percentage of error ('RPE') is defined as,

$$RPE = 100.\frac{|Bias|}{MEANobs} \qquad (12).$$

The accuracy needed for hydrological applications is a SWE RPE lower than 15% (Vachon, 2009; Larue et al., 2017), which

is the same performance objective as the CoreH2O project and the GlobSnow2 product (Rott et al., 2010; Luojus et al., 2014). This error threshold corresponds to a RMSE of about 45 kg m$^{-2}$ for a measured average Québec snowpack about 300 kg m$^{-2}$ of SWE. The ability to accurately estimate the annual SWE maximum ($SWE_{max}$) was also studied since it is one of the most important variables for hydrological applications. It allows the amount of water stored in the snowpack before the spring snow melt to be described. To avoid extreme values, the $SWE_{max}$ is estimated as the average of the SWE for a time period of +- 2

days around the detected $SWE_{max}$.

Comparing punctual data against model cells involves uncertainty due to spatial variations of the snowpack and land cover. This is a well-known problem for model validation studies and we assume here that the large number of sites and the random spatial localization of measurements within the pixels provide a useful assessment of simulations. Only snowpacks with a





SWE higher than 48 kg m$^{-2}$ (about 20 cm of snow depth), derived from measurements, were used for model evaluation to attenuate problems of shallow snow cover variability or heterogeneity.

## 4 Results

### 4.1 Simulations of $T_{B\ TOA}$

5   For each iteration of couple value ($\omega$, $\sigma_s$) (constant in frequency, Sect. 3.3.3), a couple of ($\eta_v$, $\beta_v$) values was calibrated at each frequency (11, 19 and 37 GHz) in V-pol according to the daily LAI. $T_{B\ TOA}$ are simulated from 2012 to 2016 and optimizations are performed over the summer period (see Sect. 3.3) for each site independently. The inversion is not very sensitive to $\sigma_{s.}$ (not shown) and Figure 3 shows the optimal overall $T_{B\ TOA}$ RMSE between simulated and measured $T_{B\ TOA}$ for the 12 sites and for the summer period according to the $\omega$ values. Over the summer period, a $\omega$ value at 0.07 and a $\sigma_s$ value at 0.2 cm give best

10  result for $T_{B\ TOA}$ simulations, with a minimum overall RMSE equal to 9.0 K. For this optimal ($\omega$, $\sigma_s$) couple, the mean optimal values of the $\eta_v$ and $\beta_v$ factors are detailed in Table 3 (for unfrozen ground). If the soil is frozen, the soil contribution is constant and the ($\sigma_s$, $\beta_v$) soil parameters are given in Table 3. They were previously optimized over the same study area by Montpetit et al. (2017). A value of $\omega$=0.07 is coherent with the literature for dense boreal forest areas (Pellarin et al., 2006; Meissner and Wentz, 2010; Roy et al., 2012)

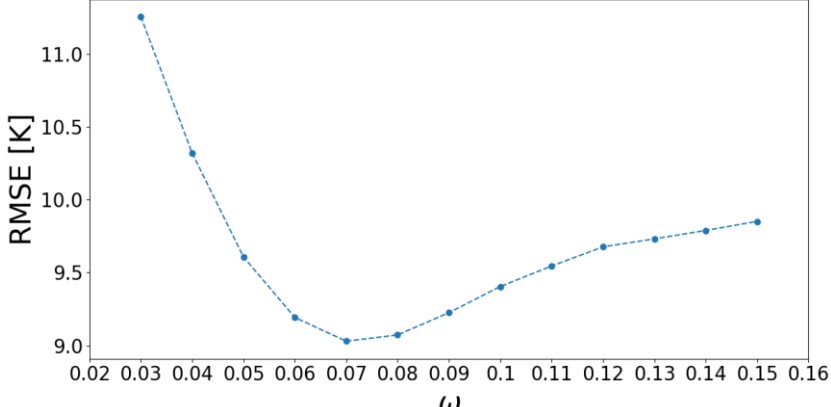

**Figure 3. Overall $T_B$ RMSE (at 11, 19 and 37 GHz, for the 12 sites and for the summer period) between the simulated and measured $T_{B\ TOA}$ as a function of the values of $\omega$. A $\sigma_s$ value at 0.2 cm gives the best results but $T_B$ RMSE is not very sensitive to this variable. The parameters $\beta_v$ and $\eta_v$ were optimized for each ($\omega$, $\sigma_s$) couple according to the frequency used.**




**Table 3. Effective parameters calibrated for the 12 studied sites to quantify soil contributions e$_{surf}$ (calibrated surface roughness 'cal. $\sigma_s$' and calibrated polarization ratio 'cal. $\beta_v$') and vegetation contributions (controlled by the calibrated ($\omega$, $\eta_v$) parameters 'cal. $\omega$' and 'cal. $\eta_v$' according to the daily LAI) measured at the top of the atmosphere. The parameterization of frozen ground was estimated by Montpetit et al. (2017). $\varepsilon_{eff}$ is the effective dielectric constant estimated with the permittivity of frozen and unfrozen soils derived from the Dobson's equations (1985). Annual and seasonal T$_{B\ TOA}$ RMSE estimated for the summer and the winter period (RMSE$_{summer}$ and RMSE$_{winter}$) are calculated from 2012 to 2016 with the calibrated parameters.**

| Frequency (GHz) | Frozen soil | | | Unfrozen soil | | Cal. $\omega$ | Cal. $\eta_v$ | Mean RMSE$_{summer}$ (K) | Mean RMSE$_{winter}$ (K) | Mean annual RMSE (K) |
|---|---|---|---|---|---|---|---|---|---|---|
| | $\varepsilon_{eff}$ | $\sigma_s$ (cm) | $\beta_v$ | Cal. $\sigma_s$ (cm) | Cal. $\beta_v$ | | | | | |
| 11 | 3.18-0.006134i | | 1.08 | | 0.69 | | 0.01 | 8.6 | 7.6 | 8.5 |
| 19 | 3.42-0.00508i | 0.19 | 0.72 | 0.2 | 0.60 | 0.07 | 0.05 | 8.7 | 9.1 | 9.1 |
| 37 | 4.47-0.32643i | | 0.42 | | 0.67 | | 0.23 | 10.1 | 35.2 | 26.8 |

Without optimizations, the annual mean RMSE of the original T$_{Bs}$ simulations varies from 12.9-47.1 K for the three frequencies (not shown). With optimizations, for the summer period, the three frequencies have a similar T$_B$ RMSE$_{summer}$ (8.6-10.1 K,

Table 3) while over the winter period the T$_{B\ TOA}$ RMSE$_{winter}$ significantly increases at 37 GHz due to the presence of the snowpack (7.6-35.2 K). The calibrations make it possible to reduce the T$_{B,37}$ RMSE by 12 K. Figures 4a, 4b and 4c show the pluri-annual T$_{B\ TOA}$ variations for Sites 12, 1 and 9, respectively, from 2012 to 2016 and at 37 GHz. At this frequency, the simulated T$_{B\ TOA}$ is strongly underestimated when a snowpack is observed. This is likely due to an overestimation of the SWE or snow grain sizes since T$_{B,\ 37}$ are attenuated in the snowpack as snow grains act as diffusers while the T$_{B,\ 19}$ and T$_{B,\ 11}$ are

relatively not affected by snow grains (RMSE$_{summer}$ similar to RMSE$_{winter}$ at 11 and 19 GHz, Table 2). Simulated SWE were overestimated by 16% and 20.2% compared to SWE measurements for Sites 1 and 9, respectively, for the winter 2014-2015. The objective of T$_B$ assimilation is to reduce these overestimations. Note that the SWE simulated at Site 12 is underestimated by 19%. The underestimation of T$_{B,\ 37}$ can also be caused by an underestimation of the vegetation contributions. This aspect is further discussed in Sect. 5.2.

By integrating ILs within the snowpack when the PR19 is above 0.015, the overall T$_{B\ TOA}$ RMSE at 37 GHz is reduced during the winter period and goes from 38.5 K to 35.2 K.

In winter, the overall T$_{B\ TOA}$ RMSE (all frequencies) is equal to 18.0 K from 2012 to 2016, similar to the overall RMSE estimated for the $\tau_{snow}$-calibrated DMRT-ML driven by in situ measurements in an open area and equal to 19.9 K compared to surface-based radiometric measurements in Québec (Larue et al., 2018).





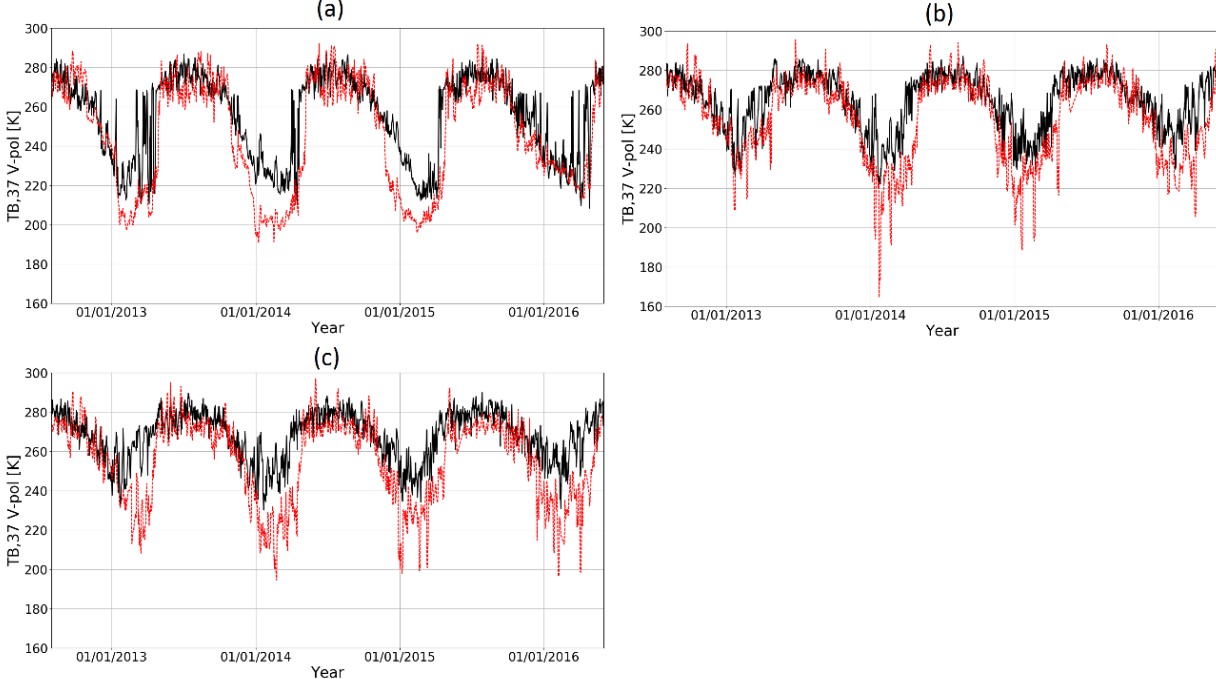

**Figure 4. Pluri-annual variations of simulated T_B TOA (red dotted lines) and measured T_B TOA (black full lines) from 2012 to 2016 at 37 GHz in vertical polarization: (a) Site 12 ($f_{cover}$ of 24%); (b) Site 1 ($f_{cover}$ of 64%); (c) Site 9 ($f_{cover}$ of 84%).**

## 4.2 Results of AMSR-2 data assimilation (DA)

### 4.2.1 Data assimilation experiments

Three DA scenarios were first tested on three sites (Site 12 ($f_{cover}$ of 24.2%), Site 1 ($f_{cover}$ of 63.7%) and Site 9 ($f_{cover}$ of 84.0%).) to determine the optimal data to assimilate: 1) both the $\Delta T_{B,19-37}$ and $\Delta T_{B,19-11}$; 2) $\Delta T_{B19-37}$ only; 3) and the three $T_{Bs}$ at 11, 19 and 37 GHz in V-pol. Figure 5 shows the daily variations of the SWE ensemble RMSE (see Eq. (11)) obtained without and with DA (prior and posterior estimates) according to the combination of frequencies used as observation. Table 4 summarizes these averaged SWE ensemble RMSEs according to the studied period (dry snow period and whole winter) for tested site.

Over these three sites and for the dry snow period, the DA reduced the overall SWE RMSE by 38.0%, 49.1% and 56.8% with scenarios 1, 2 and 3, respectively, compared to the SWE RMSE obtained with prior estimates (Table 4). The assimilation of the three frequencies helps to improve SWE simulations, giving the lowest RMSE compared to other scenarios. The same trend is observed over the whole winter and the assimilation of the three frequencies reduces the overall SWE ensemble RMSE by 45.6% (SWE ensemble RMSE of 22.7 kg m$^{-2}$) compared to the SWE ensemble RMSE of prior estimates (SWE ensemble RMSE of 41.7 kg m$^{-2}$).

In our previous work (Larue et al., 2018), we have shown a reduction of 82% of the SWE ensemble RMSE by assimilating both the $\Delta T_{B,19-37}$ and $\Delta T_{B,19-11}$ and using synthetic observation data over a dry snow period. The differences between results





using synthetic and real data in DA experiments are likely due to two aspects. Firstly, the snow model does not resolve the intra-pixel surface variability. We assumed homogeneous snow cover within the pixel in open areas, thus with no interactions between snow and vegetation. Even if we compare simulations with surface-based measurements in open areas, this could introduce large uncertainties (Roy et al., 2016). Secondly, the land cover variability and heterogeneity within each pixel also

5    induce uncertainties in the mean $T_B$ simulation over a pixel ($T_B$ weighted by the fraction of forest cover, see Eq. (2)).

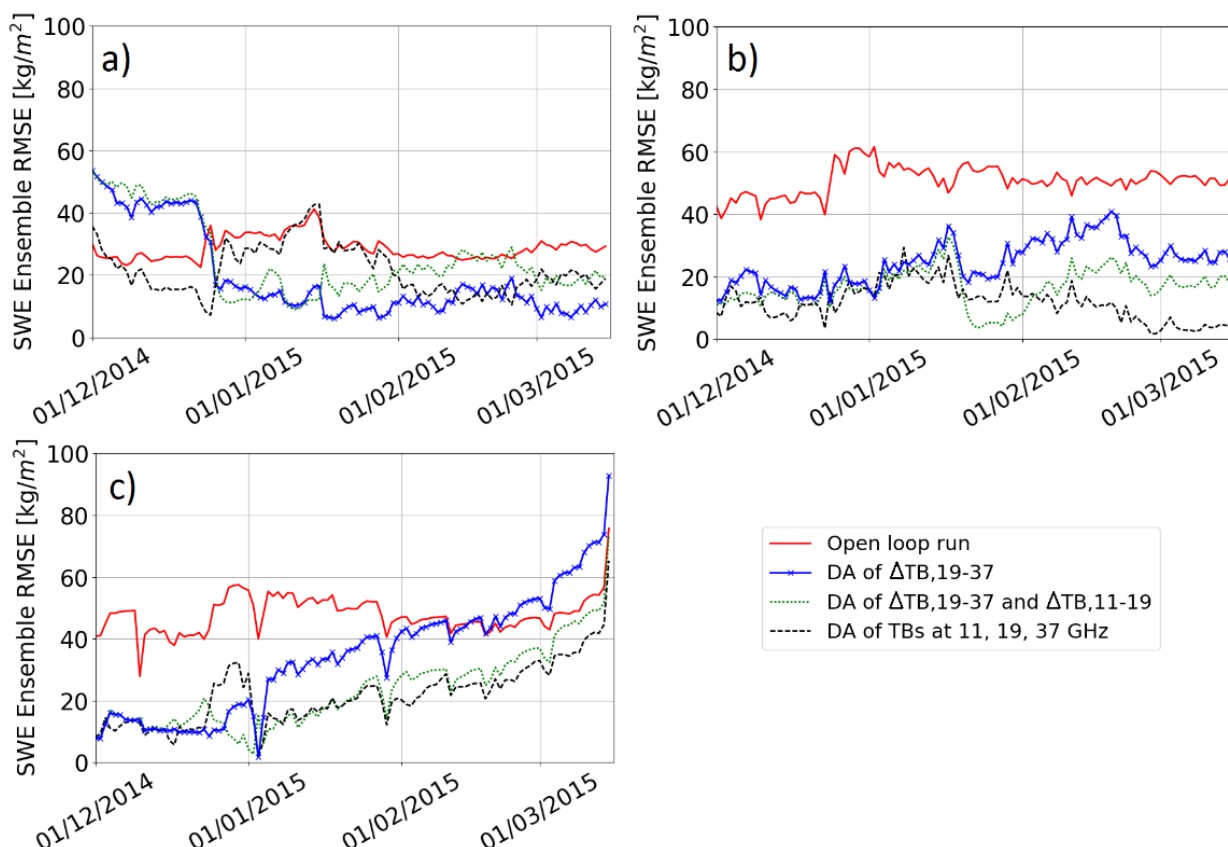

**Figure 5. Variations of the SWE ensemble RMSE (Eq. (11)) obtained with and without DA for the dry snow period (from 1 December to 15 Marsh). The red line is the SWE ensemble RMSE obtained without DA (open loop runs), the blue line is the RMSE obtained with the DA of $\Delta T_{Bv,19-37}$ only, the green dashed line the RMSE with the DA of $\Delta T_{Bv, 19-37}$ and $\Delta T_{Bv,19-11}$, and the black dotted line the RMSE with the DA of the three $T_{Bs}$. Experiments are performed for (a) Site 12; (b) Site 1; (c) Site 9, over the winter 2014-2015.**



**Table 4. Averaged SWE ensemble RMSE (see Eq. (11)) obtained with and without DA, according to the data assimilated (see Sect. 4.2.1) for each tested site. RMSE$_{dry-snow}$ is the SWE ensemble RMSE obtained from 1 December to 15 Marsh. RMSE$_{annual}$ is estimated over the whole winter (when snowpack is detected).**

| Tested sites | | #1 | #12 | #9 | Overall |
|---|---|---|---|---|---|
| RMSE$_{dry\ snow}$ (kg m$^{-2}$) | Without assimilation | 50.7 | 28.6 | 47.8 | 42.4 |
| | DA of $\Delta T_{B,\ 19-37}$ | 24.4 | 19.2 | 34.1 | 25.9 |
| | DA of $\Delta T_{B,\ 19-37}$ and $\Delta T_{B,\ 11-19}$ | 16.4 | 25.1 | 23.3 | 21.6 |
| | DA of $T_{Bs}$ at 11, 19 and 37 GHz (V-pol) | 11.8 | 21.5 | 21.7 | 18.3 |
| RMSE$_{annual}$ (kg m$^{-2}$) | Without assimilation | 47.2 | 28.9 | 48.9 | 41.7 |
| | DA of $\Delta T_{B,\ 19-37}$ | 24.2 | 23.3 | 42.4 | 30.0 |
| | DA of $\Delta T_{B,\ 19-37}$ and $\Delta T_{B,\ 11-19}$ | 18.5 | 28.2 | 31.4 | 26.0 |
| | DA of $T_{Bs}$ at 11, 19 and 37 GHz (V-pol) | 15.5 | 23.3 | 29.3 | 22.7 |

Figure 6 illustrates the comparison between SWE measurements, the original SWE Crocus simulations (SWE$_{Crocus}$) and the median of the SWE ensemble obtained with the DA of the three frequencies (referred to as 'SWE$_{DA}$'). The yellow envelope illustrates the SWE ensemble obtained without DA (prior estimates) and shows a large ensemble spread in response to meteorological forcing uncertainties. The gray envelope is the resampled SWE ensemble (posterior estimates). SWE simulations are very sensitive to the uncertainties of meteorological forcing data at the beginning of the winter season. If an

event (melting or precipitation) is missed, a constant bias on SWE estimates is kept throughout the winter. For Sites 1 and 9, the DA scheme allows the correction of these uncertainties at the beginning of the season. The SWE ensemble RMSE of posterior estimates are reduced by about 30 kg m$^{-2}$ at the beginning of the season, compared to the RMSE of prior estimates (Fig. 5). For these two sites, the SWE ensemble RMSE obtained with the DA of the $\Delta T_{B19-37}$ only increases as the snowpack becomes deeper, especially from mid-January when the snowpack becomes deeper than 100 kg m$^{-2}$ (Fig. 6). The PMW signal

from the snowpack at 37 GHz saturates for such deep snowpack (Mätzler et al., 1982; Mätzler, 1994; De Sève et al., 1997; 2007) and the assimilation of $\Delta T_{Bv,19-37}$ only does not give enough information to significantly improve SWE retrievals. For Site 9, posterior estimates are deteriorated at the end of the season compared to prior estimates with the DA of $\Delta T_{Bv,19-37}$. By adding $\Delta T_{Bv,19-11}$, this effect is reduced but stays sensitive to the depth of the snowpack (Fig. 5).

Note that the gray envelope does not always include the observations (Fig. 6a and 6c). This could be due to an under-estimation

of the R matrix. In the developed approach, the inflation technique of the R matrix is limited by a threshold on the α factor fixed at 5 since the simulations are limited by the simplifications of physical parameters in the models and we may introduce a bias if we force them to follow the observation by perturbing meteorological forcings only. Further work is needed to quantify the model errors in order to consider it in the DA scheme and to improve the representativeness of the simulations. To represent the uncertainties about the physical processes simulated with the Crocus snow model, a new system based on snow





model ensembles could be an alternative. Such an approach was recently developed by implementing different configurations estimating the physical parameters of the Crocus snow model (ESCROC, Lafaysse et al., 2017).

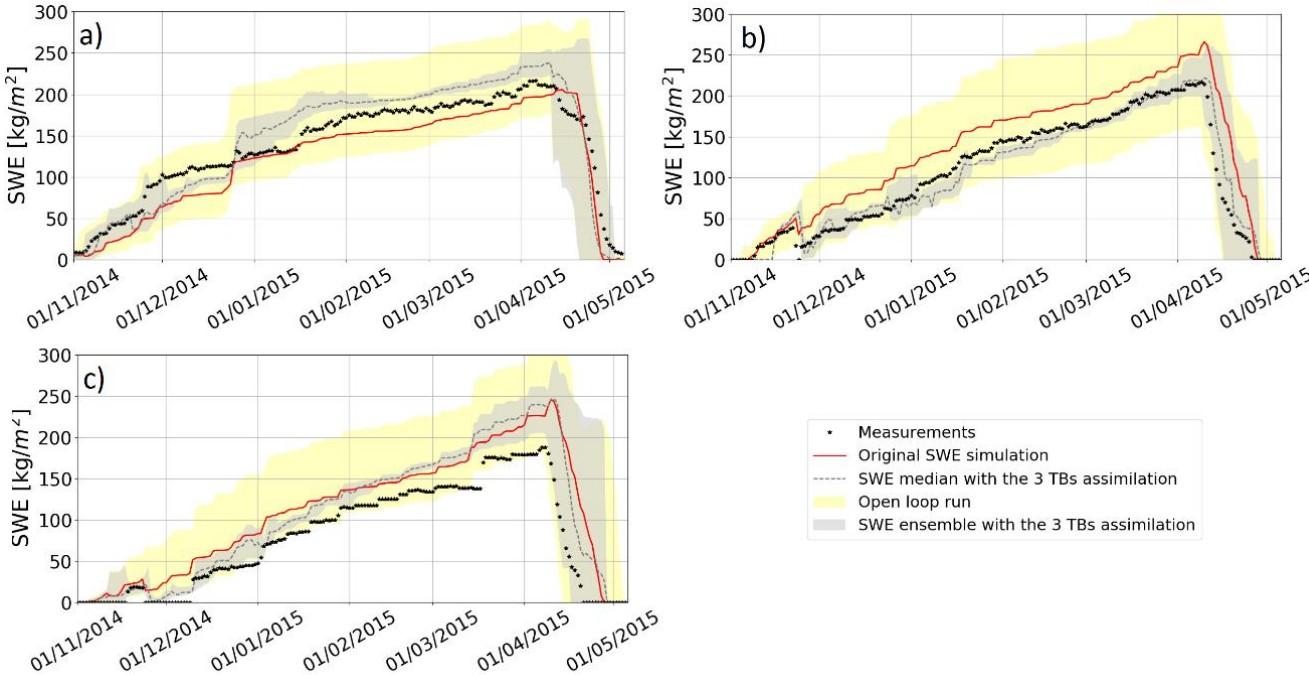

**Figure 6. Evolution of SWE measurements (black points) and SWE simulations. The SWE_Crocus is the red line and the SWE_DA is the gray dotted line. The yellow envelope is the spread of the SWE ensemble obtained with open loop runs (prior estimates). The gray envelope is the spread of the SWE ensemble obtained with the assimilation of the three frequencies (posterior estimates). Both spreads are delimited by the 5th and the 95th percentiles. Experiments are computed for (a) Site 12, (b) Site 1, (c) Site 9, over the winter 2014-2015.**

### 4.2.2 Results of T_B assimilation using the three frequencies

The median of the resampled ensemble of SWE obtained with the DA of the three frequencies (SWE_DA) is used to estimate the global performance of the DA scheme for SWE improvements. Performance is estimated for SWE up to 48 kg m$^{-2}$ (Sect. 3.4.2). Table 5 details the statistical performance of simulated SWE_DA compared to measurements and to the original SWE Crocus simulations (SWE_Crocus) over the 12 studied sites from 2012 to 2016. To analyse the impact of the vegetation,
results are separated according to the fraction of $f_{cover}$ (Table 5): moderate $f_{cover}$ ($f_{cover}$<75%, 10 sites) and high $f_{cover}$ ($f_{cover}$ > 75%, 2 sites) (see Table 2 for $f_{cover}$ site information). Figure 7 compares the SWE_DA, SWE_Crocus and SWE measurements (SWE_obs) from 2012 to 2016 for four sites with different $f_{cover}$ taken as an example: Site 5 ($f_{cover}$ = 31.5%), 10 ($f_{cover}$ = 61.8%), 1 ($f_{cover}$ = 63.7%) and 9 ($f_{cover}$ = 84.0%). In this section, we first analyze the overall SWE improvements obtained with T_B assimilation compared to original simulations. To have an idea of the impact of the DA scheme, the mean bias of SWE and



$SWE_{max}$ retrieval obtained without and with assimilation are compared and the impact of the vegetation on the quality of the DA scheme is discussed.

- **Overall SWE improvements compared to original Crocus simulations**

The overall $SWE_{Crocus}$ RMSE, bias and RPE are of 45.0 kg m$^{-2}$, 23.6 kg m$^{-2}$ and 22.1%, respectively (Table 5). In comparison,
the overall $SWE_{DA}$ RMSE, bias and RPE are improved and equal to 43.1 kg m$^{-2}$, 6.9 kg m$^{-2}$ and 18.5%, respectively. The overall bias is reduced by 17 kg m$^{-2}$ (72% of $SWE_{Crocus}$ bias) with the DA scheme. The DA of the three frequencies thus helps to improve SWE estimates over Québec. Moreover, the correlation between $SWE_{DA}$ simulations with SWE measurements gives a coefficient r of 0.79 and an offset of 10, better than those obtained with $SWE_{Crocus}$ simulations (r = 0.78, offset = 29). We analysed the number of cases with significant improvements for the total of 43 simulations studied (10 sites from 2012 to
2016, Site 11 from 2015 to 2016 and Site 12 from 2014 to 2016) by considering a 5% threshold on the bias and RMSE differences before and after assimilation. The $SWE_{DA}$ bias is significantly reduced for 25 winters (58% of cases) compared to original SWE simulations. However, the RMSE is significantly improved for only 26% of simulations, and in 49% of cases, RMSEs are similar.

- **Evaluation of $SWE_{max}$ performances**

The mean observed $SWE_{max}$ is equal to 235.6 kg m$^{-2}$ from 2012 to 2016, and the mean simulated $SWE_{max}$ is equal to 278.3 kg m$^{-2}$ and 264.6 kg m$^{-2}$ without and with the assimilation, respectively. Compared to original SWE simulations, the DA scheme improves 62% of $SWE_{max}$ simulations with an overall improvement of 13.6 kg m$^{-2}$, corresponding to 9% of SWE measurements (Table 5). Such an uncertainty extended over the whole territory could have a strong impact, considering that 1 mm of SWE in the LG watershed could represent $1M in hydroelectric power production (Brown & Tapsoba, 2007).

- **SWE accuracy for sites according to the $f_{cover}$**

The overall RPE obtained with the DA scheme is below 15% (RPE=14.6%) for sites with an $f_{cover}$ below 75% (Table 5), which is the accuracy required for hydrological applications (Larue et al., 2017). Hence, the accuracy of $SWE_{DA}$ retrievals, obtained without the use of any surface-based data, are very encouraging for hydrological needs in remote areas. In comparison, the GlobSnow-2 SWE product (Takala et al., 2011), which assimilates both $T_{Bs}$ and in situ snow depths, has a SWE RMSE equal
to 94.1 ± 20.3 kg m$^{-2}$ over the same area in Québec (Larue et al., 2017), twice the uncertainty of $SWE_{DA}$. Figures 7a and 7b (Sites 5 and 1) show that for a single site original $SWE_{Crocus}$ simulation works well for some years but can be underestimated or overestimated over other years. The DA scheme allows a more stable solution when the overall $f_{cover}$ is under 75% (not the case for Site 9, for example).

Nevertheless, even if the overall RMSE is improved, the DA scheme does not help to improve SWE estimates for sites with
an $f_{cover}$ above 75% (RMSE of 58.8 kg m$^{-2}$) compared to original SWE simulations (RMSE of 55.8 kg m$^{-2}$). The presence of vegetation is a major source of uncertainty in $T_{B\ TOA}$ simulations. The emission of the trees is superimposed on the signal emitted by the underlying snowpack and increases the $T_B$ measured at the satellite level (Chang et al., 1996, Brown et al.,





2003). At same time, the canopy also attenuates the surface emission toward the satellite. These contributions are complex to quantify since it depends not only on the tree fraction within the pixel but also on the tree species and states which emit/attenuate a different PMW signal depending on their biomass (liquid water content), volume and structure (stem, leaf, trunk) (Franklin, 1987). Also, the presence of trees modifies snow accumulation on the ground, depending on interception,

shade and sublimation effects (Dutra et al., 2011, Wang et al., 2009), which increases the spatial variability of the snowpack within the same pixel. These interactions between the vegetation and the snowpack are not taken into account with Crocus and it might induce uncertainties due to model errors. Note that SWE sensors are mostly installed in clearings, which reduces this impact in comparisons against surface-based measurements.

Kwon et al. (2016) used a similar snow radiance assimilation system to correct SD by updating the Community Land Model,

version 4 (CLM4), snow/soil states and radiative transfer model (RTM) with the assimilation of the 19 and 37 GHz of AMSR-E. Over North America, it produced significant improvements of SD for tundra type, but also produced degradations for taiga snow class and forest land cover (7.1% and 7.3% degradations, respectively). In the present study, the use of a multi-layer snowpack model makes it possible to well represent PMW emission from the snowpack with DMRT-ML, and to improve overall snowpack simulations with $T_B$ assimilation in boreal areas when the $f_{cover}$ is below 75%. Kwon et al. (2017) obtained

better results for areas with a high $f_{cover}$ in comparison to their previous study (Kwon et al., 2016) over North America by using the vegetation parameter $\omega$ as a free variable in the DA scheme, instead of pre-calibrating it as we chose to do. This aspect is further discussed in Sect. 5.2.

**Table 5. Averaged SWE RMSE, bias and RPE (Eq. (12)) over the 12 studied sites from 2012 to 2016 for original SWE simulation**
**(SWE$_{Crocus}$) and assimilated SWE$_{DA}$. Statistical performances were estimated for SWE$_{obs}$ > 48 kg m$^{-2}$ (snow depth higher than ~20 cm). $\overline{SWEobs}$ and $\overline{SWEsim}$ are the averaged observed and simulated SWE, respectively.**

|  | $\overline{SWEobs}$ | SWE$_{Crocus}$ | | | | SWE$_{DA}$ with the DA of the three frequencies | | | |
|---|---|---|---|---|---|---|---|---|---|
|  |  | RMSE (kg.m$^{-2}$) | Bias (kg.m$^{-2}$) | RPE (%) | $\overline{SWEsim}$ | RMSE (kg.m$^{-2}$) | Bias (kg.m$^{-2}$) | RPE (%) | $\overline{SWEsim}$ |
| $f_{cover}$<75% | *164.4* | 42.5 | 19.4 | 19.5 | 183.8 | 39.5 | 0.7 | 14.6 | 165.1 |
| $f_{cover}$>75% | *126.3* | 55.8 | 42.2 | 33.3 | 168.5 | 58.8 | 33.7 | 35.8 | 159.9 |
| Mean | 157.3 | 45.0 | 23.7 | 22.1 | 181.0 | 43.1 | 6.9 | 18.5 | 164.2 |





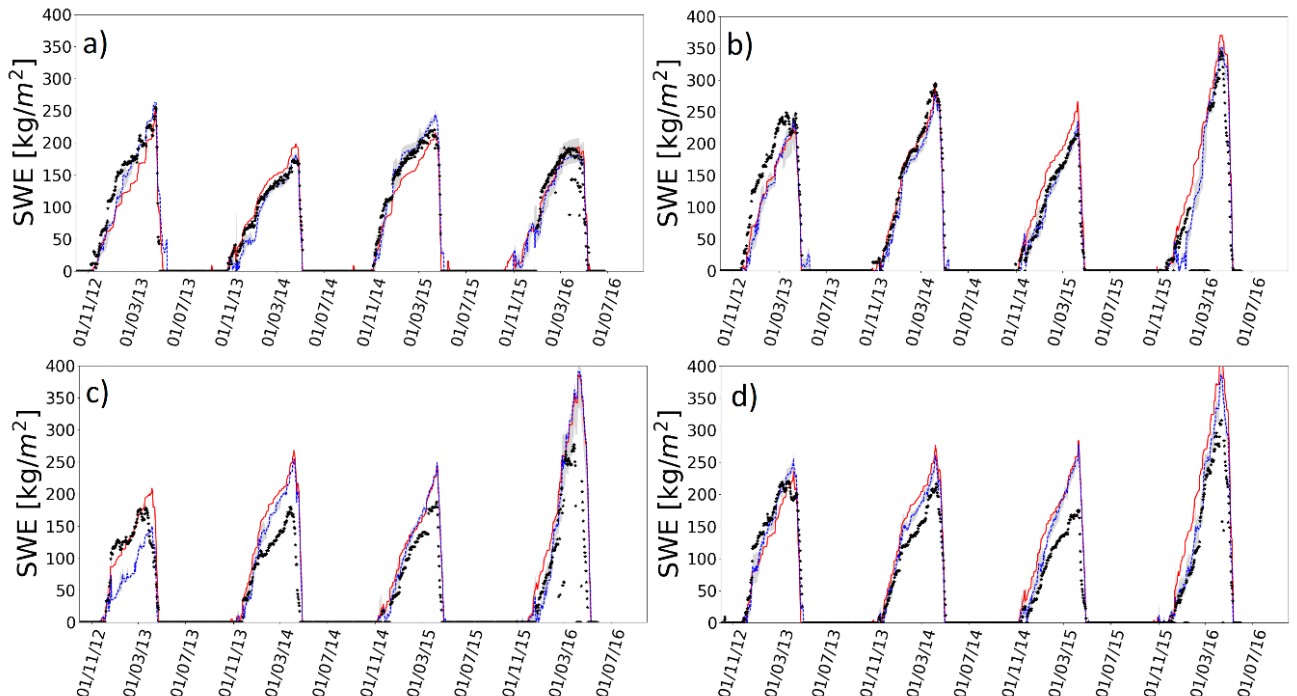

**Figure 7. Evolution of SWE measurements (black points), original SWE simulations (red full line), and the median of the SWE ensemble obtained with the DA of the three frequencies (SWE$_{DA}$) (blue dotted line). The gray envelope is the spread of the SWE$_{DA}$ ensemble (posterior estimates). Experiments are computed for (a) Site 5 ($f_{cover}$ = 31.5%), (b) Site 1 ($f_{cover}$ = 63.7%), (c) Site 9 ($f_{cover}$ = 84%), d) Site 10 ($f_{cover}$ = 61.8%), from 2012 to 2016.**

## 5 Discussion

In this section, we discuss a) the sensitivity of wet snow conditions for T$_B$ assimilation, b) the impact of using the forest parameter ω and snow microstructure (snow stickiness parameter τ$_{snow}$) as free variables in the DA scheme, and c) the percentage of surface, vegetation and atmospheric contributions in the PMW signal measured by satellite sensors.

### 5.1 Wet snow conditions

In wet snow conditions, water droplets act as emission sources (especially at frequencies above 30 GHz), and the snowpack becomes close to a black body (Brucker et al., 2011; Picard et al., 2013; Klehmet et al., 2013). The PMW observations are thus complex to use for SWE retrievals, especially at the end of the season before the spring snow melt when the SWE is maximal. Figure 8 illustrates the SWE$_{DA}$ obtained with the DA of the three frequencies applied over the whole winter and when the snow is dry only (LWC=0 kg m$^{-2}$), for Site 3 (winter 2013/2014). SWE estimates are strongly deteriorated when T$_B$ assimilation is performed in wet snow conditions. For this example, the SWE$_{DA}$ RMSE is equal to 31.1 kg m$^{-2}$ with a DA



performed over the dry snow period only and significantly increases to 70.2 kg m$^{-2}$ by assimilating $T_{Bs}$ over the whole winter (dry and wet snow conditions).

Here we used the Liquid Water Content (LWC) simulated by the Crocus model to detect wet and dry snow. This variable is subject to model errors and is linked to the original atmospheric forcing data. Further studies are needed to automatically detect

wet snow events by using direct satellite observations. Previous studies have shown the potential to use the gradient ratio ($GR=T_{B,37}-T_{B,19}/T_{B,37}+T_{B,19}$) to detect Rain-on-snow events in arctic areas (Langlois et al., 2016; Dolant et al., 2017) and this approach should be investigated for boreal forest areas in further work to improve the quality of the DA scheme for SWE improvements.  The use of active microwave observations is also a promising approach with a good spatial resolution (Roy et al., 2010).

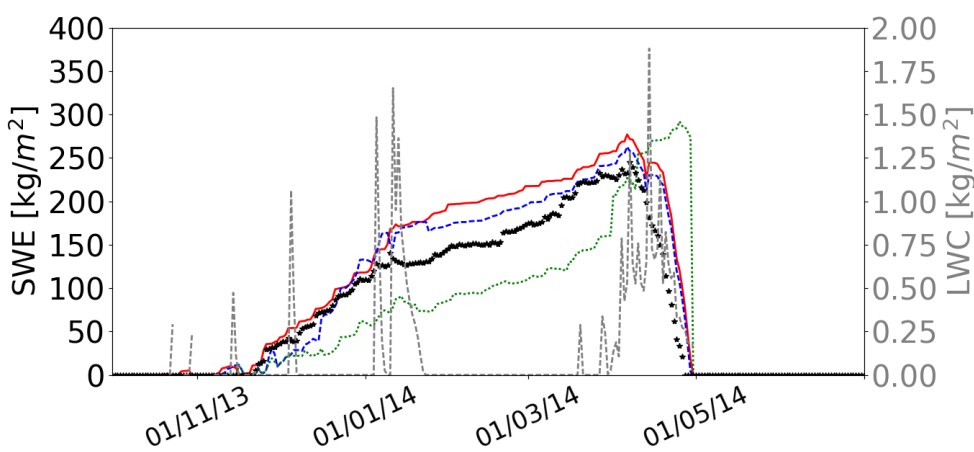

**Figure 8. Evolutions of measured SWE (black points) for Site 3 from 2013 to 2014, original SWE Crocus simulation (red full line), and SWE$_{DA}$ obtained with a DA of the three frequencies applied for the entire winter (green dotted line) and when LWC=0 only (blue full line). The simulated total Liquid Water Content (LWC) in the snowpack (dotted gray lines) is also shown.**

**5.2 Snow stickiness (τsnow) and forest parameter ω as free variables**

The quality of the PMW DA scheme could strongly depend upon the choice of the state variables. In this study, we chose to pre-calibrate forest and soil parameters and to use a constant snow stickiness parameter ($\tau_{snow}$) fixed at 0.17 (Larue et al., 2018). The forest parameter ω strongly affects the PMW emission from the vegetation, which can represent more than 60% of the signal measured by satellite sensors (see Sect. 5.3).  Kwon et al. (2017) has shown that the contribution of $T_{B\ Veg}$ to $T_{B\ TOA}$ was

better represented by considering ω in the DA scheme, and improvements in the resulting SD were evident for the forest land-cover type (about 5% with DMRT-ML). Table 6 shows the statistical performances of SWE$_{DA}$ obtained by considering ω and $\tau_{snow}$ as free variables in our DA scheme ('SWE$_{DA,\ \omega,\ \tau s}$') over the 12 studied sites from 2012 to 2016. The ω parameter was perturbed with Gaussian noise, centered on 0.07 (as calibrated) with a standard deviation of 0.02 and bounded by 0.05 and 0.12 (reasonable range of $T_{B\ TOA}$ RMSE values, Fig. 3). The snow stickiness parameter was perturbed by Gaussian noise,

centered on 0.17, with a standard deviation of 0.15 and bounded by 0.1 and 0.46. These limits correspond to the range of $\tau_{snow}$



values extracted from Larue et al. (2018) over the same study area. The ensemble size was kept to 150 members in the DA experiment.

**Table 6. Same as Table 5 but using the forest parameter ω and the snow stickiness parameter ($\tau_{snow}$) as free variables in the DA scheme to improve SWE retrievals (SWE$_{DA, ω, \tau s}$).**

| | $\overline{SWEobs}$ | SWE$_{DA, ω, \tau s}$ with the DA of the three frequencies | | | |
| --- | --- | --- | --- | --- | --- |
| | | RMSE (kg m$^{-2}$) | Bias (kg m$^{-2}$) | RPE (%) | $\overline{SWEsim}$ |
| $f_{cover}$<75% | 164.4 | 47.2 | -13.9 | 21.6 | 150.5 |
| $f_{cover}$>75% | 126.3 | 48.4 | 17.4 | 26.3 | 143.7 |
| Mean | 157.3 | 47.4 | -8.1 | 22.4 | 149.2 |

The overall SWE$_{DA, ω, \tau s}$ RMSE, bias and RPE are equal to 47.4 kg m$^{-2}$, -8.1 kg m$^{-2}$ and 22.4%, respectively, very close to the statistical performances of the original SWE$_{Crocus}$ simulations. The use of ω and $\tau_{snow}$ as free variables in the DA scheme does not help to improve SWE$_{Crocus}$ simulations for sites with an $f_{cover}$ below 75%, but the bias is significantly reduced for sites with an $f_{cover}$ above 75% (17.4 kg m$^{-2}$, or 11% of SWE measurements). In addition, the simulated SWE$_{max}$ is improved for 86% of the 43 simulations (37 cases), with a reduction of the SWE$_{max}$ bias of 36.2 kg m$^{-2}$ (23% of SWE measurements) compared to the original SWE$_{Crocus}$ simulation.

We chose to use pre-calibrated parameters because the parameters ω and $\tau_{snow}$ were not measurable and could not be directly validated. Furthermore, if we add parameters to the state variables, a larger ensemble size in the DA scheme would be needed to improve the representativeness of T$_B$ uncertainties and to ensure the solution's stability (or at least to prevent a degeneracy problem). The ensemble size was kept to 150 here but this DA experiment should produce improved results with a larger ensemble size. Nevertheless, this would require a significant computational effort. This study is a preliminary step of a PMW DA implementation for operational hydrological applications, so there was a need to limit computing time. These results suggest that the developed approach using pre-calibrated ω and $\tau_{snow}$ parameters helps to improve the retrievals for sites with an $f_{cover}$ below 75%, and the use of ω and $\tau_{snow}$ parameters as free variables in the DA scheme should be investigated in further work for sites with more than 75% forest cover.

### 5.3 Land cover contributions within the simulated T$_{B\ TOA}$

To properly assimilate PMW satellite observations, all contributions that affect the observed signal need to be well identified and quantified. The estimation of T$_{B\ TOA}$ (see Eq. (5) and (6)) can be written as the sum of the PMW contributions of the open surface (T$_{B\ surf}$), vegetation (T$_{B\ veg}$) and atmosphere (T$_{B\ atm}$) according to the fraction of forest ($f_{cover}$, estimated with the LAI as in Eq. (2) and (3)) and open area (1 - $f_{cover}$) with the Eq. (13), (14) and (15) as,

$$T_{B\ veg} = f_{cover}.\left[(1 - ω).(1 - \gamma_v).T_{veg} + \gamma_v.\left(1 - e_{surf}\right).(1 - ω).(1 - \gamma_v).T_{veg}\right].\gamma_{atm} \quad (13)$$

$$T_{B\ surf} = f_{cover}.\left[\gamma_v.e_{surf}.T_{surf}\right].\gamma_{atm} + (1 - f_{cover}).\left[e_{surf}.T_{surf}\right].\gamma_{atm} \quad (14)$$





$$T_{B\,atm} = f_{cover} \cdot \left( \left[ (1 - e_{surf}) \cdot \gamma_v{}^2 \cdot T_{B\,atm\downarrow} + (1 - \gamma_v) \cdot \omega \cdot T_{B\,atm\downarrow} \right] \cdot \gamma_{atm} + T_{B\,atm\uparrow} \right) + (1 - f_{cover}) \cdot \left( (1 - e_{surf}) \cdot T_{B\,atm\downarrow} \cdot \gamma_{atm} + T_{B\,atm\uparrow} \right)$$

(15)

Figure 9 illustrates the percentage of each contribution at 11, 19 and 37 GHz in V-pol from 2012 to 2016, for the summer and for the winter periods (defined when a snowpack is detected) for Site 12 ($f_{cover}$ of 24.2%), Site 1 ($f_{cover}$ of 63.7%) and Site 9

($f_{cover}$ of 84.0%). The percentages of each contribution are similar at 11 and 19 GHz. The contributions from the atmosphere are weak. As expected for all frequencies, the surface contributions increase for the winter period, while the vegetation contributions decrease as the LAI decreases, especially at 37 GHz. For Site 12, the surface contributions represent more than 80% of the PMW signal in winter when the vegetation contributions represent less than 10% of the PMW signal (same magnitude as atmosphere contributions). For Site 1, the surface and the vegetation contributions are similar in winter (40-55%)

whereas the vegetation contributions were more than 60% of the PMW signal in summer. For Site 9, the vegetation contributions remain the main contribution to the PMW signal in comparison to the surface contributions, even in winter (50-70% of the PMW signal for 37-10 GHz). In this dense boreal forest area, the measured snowpack emission represents less than 30% of the measured signal and SWE improvements using only $T_B$ observations is challenging. This high vegetation contribution (emission and attenuation) explain why the developed DA scheme does not succeed to significantly improve SWE

estimates for these sites with a $f_{cover}$ up to 75%.

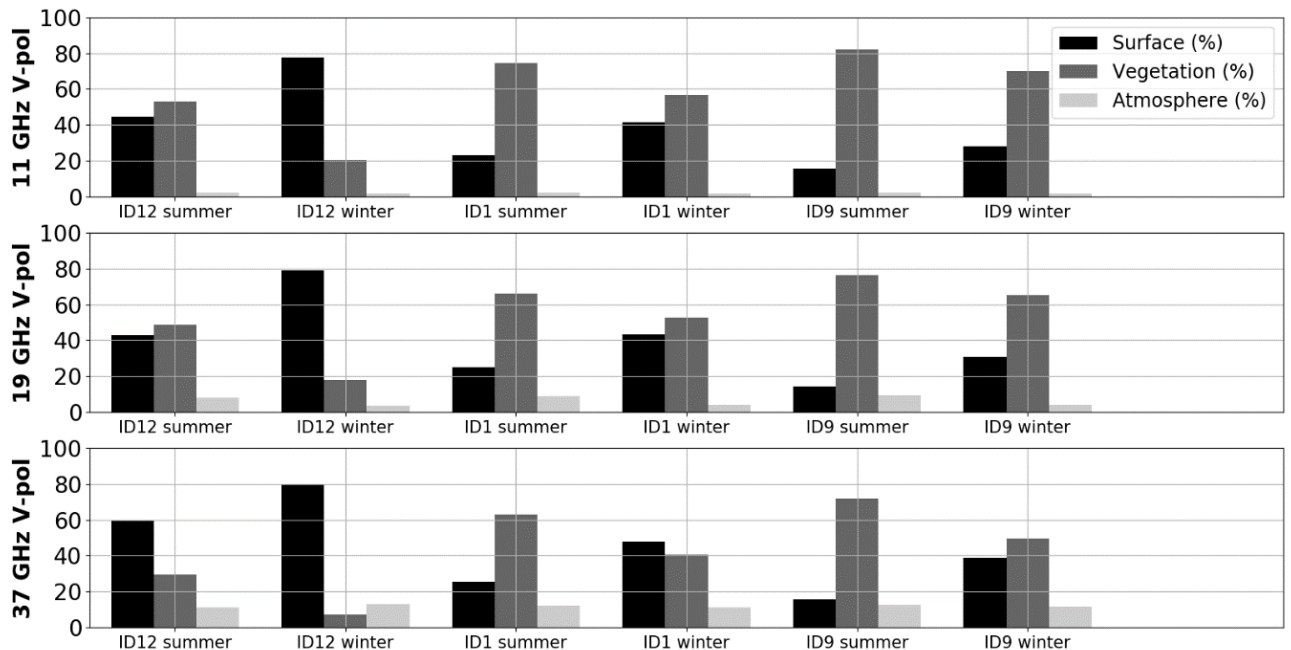

**Figure 9. Percentage of surface (black), vegetation (dark gray) and atmosphere (light gray) contributions to the simulated PMW signal at the top of the atmosphere at the three frequencies 11 (top), 19 (middle) and 37 (bottom) GHz. ID12, ID1 and ID9 are site 12 ($f_{cover}$ of 24.2%), 1 ($f_{cover}$ of 63.7%) and 9 ($f_{cover}$ of 84.0%), respectively. Summer and winter periods are defined when snowpack is**

**observed or not.**



## 6 Summary and conclusion

An ensemble data assimilation (DA) scheme was implemented in a calibrated chain of models (Crocus/DMRT-ML, soil, vegetation and atmosphere radiative transfer models) to improve SWE estimates by updating forcing data and snowpack states

with the assimilation of AMSR-2 satellite observations. The developed approach does not use any surface-based data and was tested over a boreal area in Québec (Eastern Canada). The proposed DA scheme is a particle filter with a resampled SIR algorithm, using an inflation technique of the R matrix to avoid degeneracy problems. The multi-layer snowpack model Crocus, coupled to the surface land model ISBA, was used to simulate the evolution of the snowpack. The DMRT-ML, the ($\omega$-$\tau_{opt}$) model, an atmospheric model and the WM99 radiative transfer model were calibrated to simulate the PMW contributions from

the snowpack, the vegetation and the soil, respectively, at the top of the atmosphere. The DA scheme was performed over 12 sites from 2012 to 2016, only in the presence of dry snow. Ice lenses were detected and integrated in the snowpack by using a thresholding approach on the polarization ratio at 19 GHz. The study shows:

    1-  $T_{B\ TOA}$ can be well simulated with the developed chain of models. By calibrating soil and forest parameters ($\omega$=0.07 and $\sigma_s$=0.2 cm), the overall $T_{B\ TOA}$ RMSE (all frequencies) is equal to 18.0 K from 2012 to 2016 over the winter

period. This RMSE is similar to the overall RMSE estimated for the $\tau_{snow}$-calibrated DMRT-ML driven by in situ measurements in an open area (19.9 K compared to surface-based radiometric measurements in Québec (Larue et al., 2018)).

    2-  The assimilation of $T_{Bs}$ at 11, 19 and 37 GHz (V-pol) improves the SWE retrievals compared to the assimilation of $\Delta T_{B\ 19-37}$ only (sensitive to snowpack depth) or to the assimilation of both $\Delta T_{B\ 19-37}$ and $\Delta T_{B\ 11-19}$. The SWE RMSE of

posterior estimates is reduced by 45.6% over the whole winter compared to the SWE RMSE of prior estimates (open loop runs).

    3-  By using calibrated $\omega$ and $\tau_{snow}$ parameters in the DA scheme, the overall bias (for 12 sites from 2012 to 2016) of the original $SWE_{Crocus}$ simulations is significantly reduced by assimilating $T_{Bs}$ at 11, 19 and 37 GHz (from 22.1 kg m$^{-2}$ to 6.9 kg m$^{-2}$). The bias on $SWE_{max}$ is reduced by 13.6 kg m$^{-2}$ (9% of SWE measurements). The overall RPE goes from

22.1% to 18.5%, which is close to the range of accuracy needed for hydrological applications (SWE RPE < 15%). This accuracy is achieved with the $T_B$ assimilation for sites with a $f_{cover}$ below 75%.

Even with the difficulties associated with quantifying all the different factors that contribute to the PMW signal measured by satellite sensors in remote boreal areas (canopy, ice crust, frozen ground / unfrozen, presence of lakes, moisture in the snow,

topography, etc.) (Kelly et al., 2003, Koenig & Forster, 2004), and even when vegetation contributions are 50% of the PMW signal, the implementation of a DA scheme in a well-calibrated chain of models allows to reduce SWE uncertainties without using any surface-based data. This assimilation scheme can be easily implemented in an operational system using real satellite-



borne observations, despite the relatively significant computing time required. This research opens the way for global applications to obtain more accurate SWE estimates over large and remote areas where few meteorological weather stations are present.

*Data availability.* The daily SWE data provided by Hydro-Québec are used for hydrological purposes and are not available to the public due to legal constraints on the data's availability. The SWE data, SD data and field campaign measurements provided by the University of Sherbrooke will soon be available on the GRIMP snow group website http://www.grimp.ca/data/. Meteorological    GEM    data    are    freely    available    on    the    Government    of    Canada's    website https://weather.gc.ca/grib/grib2_reg_10km_e.html. Other data used are listed in the references.

*Competing interests.* The authors declare that they have no conflict of interest.

*Acknowledgements.* The authors would like to thank the data providers: Hydro-Québec, Environment Canada (CMC-ECCC) and the University of Laval. This project was supported by financial contributions from NSERC, Canada, FRQ-NT and
MITACS Québec, and the CFQCU France-Québec collaboration program.

**Appendix A: Online adjustment of the observation error covariance matrix R**

Online adjustment of covariance matrices in data assimilation is quite a common approach with the Ensemble Kalman filter (Dee, 1995; Miyoshi, 2001, Brankart et al., 2010, 2011) but not with the particle filter. However, in many implementations of the particle filter, the measurement pdf is considered Gaussian, so that the particle weights are computed using the observation
error covariance matrix R. This matrix can therefore also be subject to adjustment in the context of the particle filter. Online adjustment can be and is often performed by tuning a simple inflation of the initial covariance matrix. This is the approach chosen here.

Noting $\delta_i = y - h(x_i)$ the innovation for particle i, the weight of this particle is

$$\widetilde{we}_i = \frac{we_i}{\sum_j we_j} \tag{A1}$$

where

$$we_i = \exp\left(-\frac{1}{2}\delta_i^T R^{-1} \delta_i\right) \tag{A2}$$

An inflation of matrix R by a factor $1/\alpha$ (larger than 1) can be interpreted as an exponent $\alpha$ (smaller than 1) to $we_i$. Because the weights ˜ wi are nonlinear functions of R, inflating R tends to flatten their distribution. Online adjustment consists in finding a value for $\alpha$ that flattens the distribution of weights to the point where $N_{keep}$ particles are selected with certainty, $N_{keep}$ being a
number to be prescribed. The number $N_{keep}$ being fixed, if the resampling step is performed using Arakawa's procedure (Arakawa, 1996), the weight of the $N_{keep\text{-th}}$ particle to be selected, $\widetilde{we}_{keep}$, must become equal to $\widetilde{we}_{ref} = 1/N_{keep}$. Consequently,





$$\widetilde{we}_{keep} = \frac{(we_{keep})^{\alpha}}{\Sigma_j (we_j)^{\alpha}} = \widetilde{we}_{ref} \tag{A3}$$

or, written differently after taking the logarithm:

$$\alpha = \left(\log(\widetilde{we}_{ref}) + log\left(\Sigma_i \left(we_j\right)^{\alpha}\right)\right) / log(we_{keep}) \tag{A4}$$

This equation for α is not solvable analytically. Instead, we find α after the convergence of the series:

$$\alpha_n = \left(\log\left(\widetilde{we}_{ref}\right) + log\left(\Sigma_i we_j^{\alpha_{n-1}}\right)\right) / log(we_{keep}) \tag{A5}$$

The result of this adjustment is illustrated in Figure A1. The blue dots show the first 20 weights of a sorted distribution for an ensemble of 50 particles strongly prone to degeneracy: only 4 particles have a weight larger than 1/50 = 0.02. The minimum number of particles to be selected is fixed to $N_{keep}$ = 10. After the adjustment procedure, the identified inflation factor for matrix R is 3.6 (α = 0.277) and the weight $we_{keep}$ of the 10th particle is exactly equal to 0.02.

Obviously, this procedure is used only if the number of selected particles is below the $N_{keep}$ threshold with the initial weights.

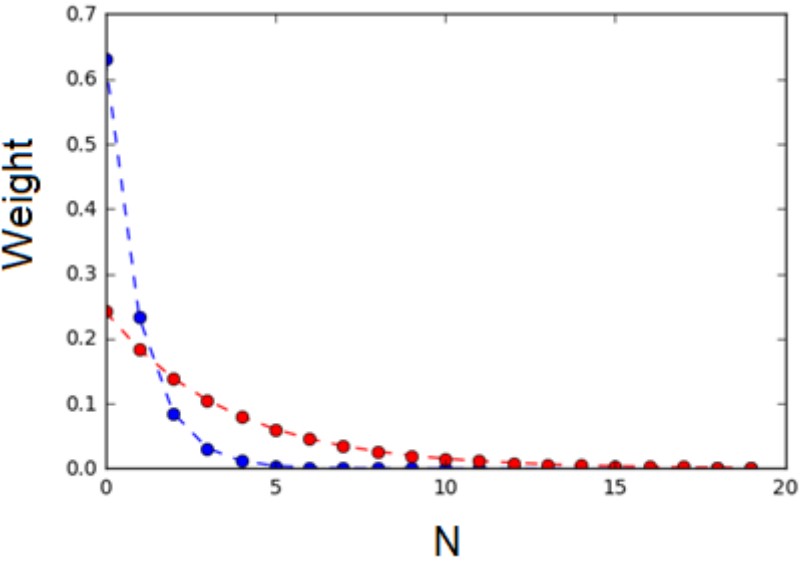

**Figure A1. Weight distribution of the first 20 weights of a sorted distribution for an ensemble of 50 particles: distribution before the adjustment (blue dotted points), showing a degeneracy problem, and distribution after the adjustment procedure (red dotted points), where weight distribution is 'flattened' and significant weights are distributed around $N_{keep}$ particles (10 particles for this example).**

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
