# Peer review of "Assimilation of passive microwave AMSR-2 satellite observations in a snowpack evolution model over North-Eastern Canada"

_Hydrology and Earth System Sciences, 2018_

## Referee Comment (RC1) · Anonymous Referee #1 · 13 Apr 2018

Review of "Assimilation of passive microwave AMSR-2 satellite observations in a snowpack evolution model over North-Eastern Canada", by Larue et al.

The authors present an excellent case study, using a particle filter to do radiance assimilation for snow for the first time in the literature. They build on a previous synthetic study, and validate at 12 sites with in situ stations in Quebec.

Overall, I highly recommend publication in HESS. This is excellent work. However, I think the presentation could be much improved. The standard of English usage is a little bit short of the HESS standard; I flagged some of the problems below, but there are many more. There are places where the symbols are undefined, or things are not

explained well. A bit more work would greatly improve some of these things.

The only other comment is that overall, the authors find (in my summary) that they run at 12 sites, and find a quite marginal improvement in RMSE: from 45 kg mˆ2 to 43.1 kg mˆ2 over all sites. Given the small sample size, those may be statistically indistinguishable. There's a lot of encouraging results too: the bias present in the open loop runs is much reduced, and is essentially zero over the eight sites with less than 75% forest cover. The authors start the results presentation with a deep dive on three sites that do quite well. They ought to give a rationale there, to avoid looking like they are "spinning" the results too much. The authors should acknowledge the small sample sizes involved; they start with 12 and then split things out into low and high forest cover, so they are looking very few sites. This is understandable, but it does mean they need to acknowledge that sample sizes are perhaps not statistically large enough to be able to make all of the claims they might want to.

Minor Comments

1. Page 2, line 16: Please also cite: Andreadis, K. M., and D. P. Lettenmaier (2012), Implications of representing snowpack stratigraphy for the assimilation of passive microwave satellite observations, Journal of Hydrometeorology, 13(5), 1493–1506, doi:10.1175/JHM-D-11-056.1. 2. Page 3, line 9-10: Please provide a recap the main findings of this previous study, especially to the extent they bear on this paper. Recommend moving page 13, lines 4-8 up to the introduction. 3. Page 5, line 13: Here and elsewhere (e.g. Page 12, line 11): Presumably Crocus is running at a 1-hour timestep, and you are outputting daily. Please clarify. 4. Page 5, line 17: "total of precipitable water". Remove "of" 5. Page 6, line 11: "the observations errors were". Grammar doesn't work here. Accepted usage should be "the observation error was" but you could also just change to "observation" and otherwise keep the same. 6. Page 7, line 2: "Database" should be "Data". 7. Page 8, line 14: "dense forested" should be "densely forested". 8. Page 8, line 15: The signal is not in this case biased. I don't think you can talk about the T_B observation being biased unless e.g. AMSR-2 is

measuring TOA values that are biased compared to true TOA values. Instead, I think you mean that it's contaminated or significantly affected by the forest. Treating the TOA measurement as if it were a measurement of $T_{B}$ just above the snow would result in a biased comparison. Anyway, please revise. 9. Page 9, line 21: Crocus has several options for computing grain size. Please give the details here of how this was done for this study, even if they are already reported in the previous Larue et al. 2018 paper. As the authors know so well, $T_B$ is more sensitive to grain size than to SWE, at least at 37 GHz. So this is a really key part of the paper. 10. Page 10, lines 11-22. I read this a few times, but am still confused. So once detected, an IL is added at the top of the snowpack. Then on the first timestep with precipitation, it is subsequently buried 4 cm beneath the surface? So e.g. it would exist in the model at the top indefinitely as long as there is no snowfall? Why not just add it 4 cm under the surface from the time it is detected? 11. Section 3.3, pages 10-12. Overall, I found the notation and presentation to be confusing enough to be distracting here. I would start out the section with an equation that includes both forest and atmosphere; it is frustrating that it starts with an equation neglecting the atmospheric contribution, given the title of the section. I also find it confusing that the atmospheric contributions are presented in a section entitled "Vegetation contributions." Please revise. 12. Page 11, line 12. I believe that "simple" should be "single", correct? 13. Page 12, line 5. What does 0.1 represent? Probably better to define as a symbol, and give the value in the text. 14. Page 12, section 3.3.2. Overall I think that you ought to be able to read the section on soil contributions and know which of the parameters are dependent on frequency, and which are frequency invariant. You'll need to revise 3.3.3 a bit too, I think, to avoid duplicating too many explanations. 15. Page 12, line 6. What is the definition of $r_H$ in equation 8? 16. Page 12, line 8. Is the "." supposed to represent multiplication? If so, please remove, and just take advantage of implied multiplication, writing e.g. $\sigma_s = k \sigma$. 17. Page 12, line 9-11. Is $\Gamma$ frequency-dependent? 18. Page 12, line 12-13. I think I see now that you are using $\nu$ to note frequency-dependent variables, and to distinguish from those that are frequency-invariant. However, it took
me a while to work this out. Can you reword this, maybe: "Note that we will often use "\nu" subscript to denote quantities that are dependent on frequency, hereafter." 19. Page 12 line 15. Sometimes the process of backing out model parameters is referred to as "calibration" and sometimes as "inversion" in this paper. Later (in the results) it's referred to as "optimizations" (Page 17, line 8-9, e.g.). Please just pick one of those two names and use it at all times, to avoid confusion. Else readers wonder if you are referring to the same thing, or to something they missed somewhere in the paper. 20. Section 3.3.3, pages 12-13. This section required far too long to parse. I found it to be unnecessarily opaque. If this is the same procedure as Roy et al. 14, I would just say that you used the same procedure as that paper. If not, can I recommend a thorough rewrite? Something like: "We thus have two frequency-dependent parameters (eta_nu, beta_nu), and two frequency-invariant parameters (omega, sigma_s). We perform a sort of two-stage calibration. We permute all possible combinations of the two frequency invariant parameters. Specifically we varied omega from 0.02 to 0.16 in steps of 0.01, and varied sigma_s from 0.01 to 1.1 in steps of 0.05. This yields a total of 300 possible combinations of the frequency invariant parameters. Then, for each possible combination of the frequency-invariant parameters, we performed a calibration of the frequency-dependent parameters, eta and beta, for each frequency; thus a total of 900 frequency-dependent calibrations are performed. Finally, for each possible combination of the frequency-invariant parameters, we compute the total post-calibration Tb_RMSE across all three frequencies. The combination of frequency-invariant parameters resulting in the lowest Tb_RMSE is chosen." 21. Page 13, lines 9-12. Is the implication that everything is identical to the previous paper except for the covariance inflation? If so, please make this explicit. If not, then no need to highlight covariance inflation prior to beginning the first subsection, in my opinion. 22. Page 14, line 14. I think ideally you'd have the observation error be larger than 2 K. It really represents all mis-match between observation and model: i.e. what error is expected if the model in its current form is run with "correct" inputs? Of course, this is only a sort of initial value, since you are using covariance inflation. May want to make that explicit here. 23. Page

14, line 22. Can you clarify that observation error covariance here is just observation standard deviation squared times the identity matrix? 24. Page 15, line 25. I don't think it is 15% for CoreH2O for shallow snow. I think the requirement was given in absolute SWE (mm) for shallow snow, and a percentage for deep snow. Please double check. 25. Page 15, line 21. What do you mean by "punctual"? Please reword. 26. Page 15, line 32. I believe there are twelve total sites. Please make that explicit. Usually you want >20 for e.g. large-sample statistics to hold, right? 27. Page 15, line 33. Why do you think the site selections are random? In the Western US mountains (albeit a very different environment), it is assumed that logistics of site selection end up leading to a highly biased spatial distribution. E.g. seeMolotch, N. P., and R. C. Bales (2005), Scaling snow observations from the point to the grid element: Implications for observation network design, Water Resources Research, 41(W11421), doi:10.1029/2005WR004229. 28. Page 16, line 4. I recommend retitling the first subsection "Results of model calibration". 29. Page 16, line 5-7. This entire first paragraph is methods. It must NOT be in the results section. Please move it to the methods section, probably §3.3.3. Also please see my suggestions for reworking §3.3.3. 30. Page 17, line 10. I thought you were not calibrating over the winter? Please clarify. Is this using the optimal parameters you obtained over the summer and combining with the open loop model run? Or are you also calibrating over the winter? Recommend describing the winter error statistics very carefully; to be honest, I think having them in there is not worth the added confusion it brings to the reader. The calibration should really be in the background, here, as it has been done in many previous papers. The focus should be on the assimilation results. 31. Page 17, line 12, and elsewhere. "Pluri-annual" is not common English usage. Please reword. 32. Page 21, line 12. What is meant by the 48 kg/mˆ2 limit? This seems to appear from nowhere, and additionally represents very shallow snow. 33. Page 22-23. Recommend simply removing these sub-section headers. They are fairly clear from context, and the sections are not too long. 34. Page 22. I don't think you can claim 0.79>0.78 without doing a rigorous statistical test; they seem basically the same to me. The offset is a definitely change; I would highlight that.

35. Page 22, line 25. Recommend giving RPE here instead / in addition, since that's what is being discussed earlier in the paragraph. 36. Page 23, Table 5. Recommend redoing notation. Why are italics used in random places? Why aren't "obs" and "sim" subscript? 37. Page 24, line 8. Recommend introducing "wet snow" as an issue in the introduction. Add some text maybe on how passive microwave won't give additional information about snow once the snow is wet, but can help correct earlier biases, etc. It is only mentioned once in the methods, and is quite easy to miss. 38. Page 25, Figure 8. Can you show the $SWE_{DA}$ posterior ensemble spread, as in the other graphs? I think we should see it get larger when liquid is present, which should enrich the discussion in this section. 39. Page 26, line 27. Do you mean "However", instead of "Nevertheless"? 40. Page 27, Figure 9. Is this for posterior or open loop?

---

## Referee Comment (RC2) · Anonymous Referee #2 · 17 Jul 2018

This paper evaluates the assimilation of AMSR-2 brightness temperature observations at 11 GHz, 19GHz and 37GHz into the Crocus/DMRT-ML models to analyse snow water equivalent. Results are evaluated against in situ data obtained from 12 sites representing different land cover types in Québec, Eastern Canada. This study is very relevant for the scientific community as assimilating radiances to analyse snow conditions in physical snowpack models is of high interest for hydrology and numerical weather prediction applications. The paper shows promising results for moderate vegetation cover and the method opens larger scale applications possibilities. The text has a number of language issues and grammar mistakes, some of which are listed below. The manuscript should be checked by a native English speaking colleague. However

the paper is very well organised, results are presented with appropriate figures and tables, and very interesting discussion and conclusion sections are provided. I suggest this paper to be accepted for publication in HESS after the comments below are accounted for.

Page 9 lines 11-12: " To generate a three hourly-continuous meteorological forcing database for running Crocus, successive GEM forecasts were taken from the +09 forecast hour to the +18 forecast hour provided at the 00 and 12 UTC analysis time of each day." It is not clear to me what it means. Did the authors take forecasts at 00 UTC steps covering 09UTC to 12UTC and at 12UTC steps covering 18UTC to 21 UTC? Does it match the AMSR2 pass at 1pm local time? The authors should clarify in the text and also it would be clearer to use UTC everywhere, also when describing the AMSR data both UTC and local time could be provided (page 5 section 2.2 and page 7 section 3.1.1).

Page 10 lines 18-20: " Hence, as soon as a snowfall is detected with GEM precipitation data, the IL firstly added on the top of the surface was positioned 4 cm from the surface in the simulated snow profile. The maximum number of detected IL was fixed at two. In this case, the first detected IL was 20 positioned at 8 cm from the surface and the second at 4 cm after a snowfall was detected." please clarify/reformulate this part. From the first sentence the reader understands that the first IL is at 4cm depth, but from the second and third sentences it is indicated that the first one is at 8cm depth and the second at 4cm.

In section 3.4.2 it would be very useful to include a table with the list of experiments with a short name for each and indicating in the caption the experiment period and sites. So that the reader would have centralised in the table the experiment set-up information. For example experiment names like "DA1_TB19-37", "DA2_TB19-37,TB11-19", "DA3_TB_11,19,37". Using these short names in Section 4 when presenting the results would be much clearer. Also the first sentence of the section 3.4.2 starts with "In a first step,...", it should be followed by "In a second step," probably page 15, line

17 when introducing the experiment on all sites. It is not a problem to reveal the fact that the best configuration will be DA of TB11,19,37 at this stage, it will only make the paper clearer. Page 18: section 4.2.1, first paragraph: there is no need to repeat in the results section the experiments that were conducted: please remove the first sentence of section 4.2.1. Finally, even the last experiment, with free snow stickiness and forest parameter, presented in Section 5, should be described in Section 3.4.2 and included in the experiment list table. It is surprising for the reader to be informed in Section 5 that another experiment was conducted. In other words, the paper should not necessarily follow the chronology of the research developments. It should present the experiments and the results/discussion without holding new experiment description for the results and discussion sections.

Page 20 line 6 and Figure 6: The way it is formulated page 20 and in the caption, it is not straightforward to understand the meaning of "SWE ensemble obtained with the DA of the three frequencies (referred to as 'SWEDA')". The reader may wonder why this one is called SWEDA whereas the DA of TB 11, 19 37 GHZ had no specific name. Please use experiment names and provide a table in section 3 (see comment above).

Minor comments:

table 1 Caption: replace: "Characteristics of the nivometric stations: SWE (in kg m-2) data, Latitude (Lat.),..." by "Characteristics of the nivometric SWE stations: Site number, Latitude (Lat.),"

Table1 caption: GEM is used here but only defined later in section 2.2. So, define it on its first occurence in Table 1's caption.

Page 5, line 11: remove "further"

Page 5 line 13: replace "1pm" by "1pm local time"

Page 5 last paragraph: make sure tenses are consistent: line 12: "Crocus computes" and line 13-14: "The DMRT-ML ... was used to "

Page 6 line 13: ".." -> "."

Page 8, table 2, caption: clarify if the winter period January to March as indicated in the caption, or if it is 1 January to 1 March as indicated in the text page 7 line 18.

Page 8 line 17 "formulations in" -> "formulations of"

Page 10 line 24: reformulate by something like: "The PMW brightness temperature (TB) emitted at the scale of the AMSR-2 product can be written as (2) for each grid cell as"

Page 11 line 15: "the expression of TB TOA in boreal areas was described by the Eq. (2)" for consistency please update Eq 2 (page 10) by replacing "TB=" by "TB TOA="

Page 12 line 3: remove "throughout the year"

Page 12 line 7: "In the Eq.7"–> "In Eq. 7"

Page 12 lines 17-20: update the text to ensure tenses consistency. For example lines 17-18: "Forest parameters ($\omega$, $\gamma\nu$) depend on the forest characteristics, such as the biomass and the structure of the canopy for each site. To take into account the temporal variations of these caracteristics, the forest parameters were linked to the LAI." can be replaced by: "Forest parameters ($\omega$, $\gamma\nu$) depend on the forest characteristics, such as the biomass and the structure of the canopy for each site. They also depend on LAI which allows to account for the seasonal cycle in the forest emission". Also check the rest of the paragraph.

Page 12 line 31: replace "value couple" by "set of values", replace "(considered constant in frequency)" by ", defined at each frequency (11 GHz, 19 GHz and 37 GHz)", and replace "for each frequency (at 11, 19 and 37 GHz) in V-pol" by "at V-pol".

Page 12 line 32 / page 13 line 1: This sentence is not clear, it should be reformulated. Do you mean that the parameters were optimised also at H-pol or that the V-pol set of parameters were tested at H-pol?

Page 13 lines 9-12: the 3 sentences on observation errors and ensemble inflation technics should be removed because they are not well formulated and the description given on the next page (page 14) is very clear.

Page 14, lines 19-21: This sentence repeats lines 12-13: " Hence, to avoid a 20 degeneracy problem, the weight of the 25-th selected particle (wekeep) must always be larger or equal to the inverse of the ensemble size (N=150)." Please update the text to avoid repeating sentences.

Page 15, first paragraph, last sentence: this statement should be placed earlier in the paragraph, before the three experiments are described.

Page 14 line 30 and page 15 line 8: The information on the DA experiments length and period should not be spread in the test. It should be clearly stated once.

Page 16 line 5: remove "(constant in frequency, Sect. 3.3.3)"

Page 16, section 4.1: update the text to use consistent tenses.

Page 18 line 19: replace "according to the studied period" by "for the studied period" Page 27 lines 14-15: replace "explain" by "explains" and "up to" by "larger than".

---

## Author Comment (AC1) · 10 Sep 2018

General comment of the R1Âă:

Review of "Assimilation of passive microwave AMSR-2 satellite observations in a snow-pack evolution model over North-Eastern Canada", by Larue et al. The authors present an excellent case study, using a particle filter to do radiance assimilation for snow for the first time in the literature. They build on a previous synthetic study, and validate at 12 sites with in situ stations in Quebec. Overall, I highly recommend publication in HESS. This is excellent work. However, I think the presentation could be much improved. The standard of English usage is a little bit short of the HESS standard;

I flagged some of the problems below, but there are many more. There are places where the symbols are undefined, or things are not explained well. A bit more work would greatly improve some of these things. The only other comment is that overall, the authors find (in my summary) that they run at 12 sites, and find a quite marginal improvement in RMSE: from 45 kg mËĘ2 to 43.1 kg mËĘ2 over all sites. Given the small sample size, those may be statistically indistinguishable. There's a lot of encouraging results too: the bias present in the openloop runs is much reduced, and is essentially zero over the eight sites with less than 75% forest cover. The authors start the results presentation with a deep dive on three sites that do quite well. They ought to give a rationale there, to avoid looking like they are "spinning" the results too much. The authors should acknowledge the small sample sizes involved; they start with 12 and then split things out into low and high forest cover, so they are looking very few sites. This is understandable, but it does mean they need to acknowledge that sample sizes are perhaps not statistically large enough to be able to make all of the claims they might want to.

General comment from the authorÂă:

It is not only 12 sites, it is 43 winters which were compared (see table 1 for time period of each station), and these winters were different enough to ensure to study a large range of snowpack observed in Québec. We added the followong line in the 'study area' section : "A total of 43 winters could thus be studied (Table 1). These winters were all very different, the winter 2012-2013 had the lowest snow accumulation in ten years (165 cm) whereas the winter 2013-2014 was very snowy (379 cm) compared to the average snow accumulation (217 cm). The winter 2014-2015 was unusually cold (3 ° below the average temperatures) and the winter 2015-2016 was the warmest in 60 years (statistics can be find at http://www.mddep.gouv.qc.ca)." The following paragraph was rewritten and moved at the begining in section 2.2 (page 6, line 26): " Comparing data simulated at the station against model cells involves uncertainty due to spatial variations of the snowpack and land cover. This is a well-known problem for model

validation studies and we assume here that the high number of sites (12 SWE stations, or 43 snowpack simulations) provides a useful assessment of simulations. It is also known that the spatial localization of measurements can lead to some biases (Molotch and Bales, 2005). To diversify its measurements, Hydro-Québec has installed two SWE sensors in the forest, and not in a clearing as is the usual practice for ease of maintenance. " Answer to minor Comments

R1. Page 2, line 16: Please also cite: Andreadis, K. M., and D. P. Lettenmaier (2012), Implications of representing snowpack stratigraphy for the assimilation of passive microwave satellite observations, Journal of Hydrometeorology, 13(5), 1493–1506, doi:10.1175/JHM-D-11-056.1.

AC1. Done

R2. Page 3, line 9-10: Please provide a recap the main findings of this previous study, especially to the extent they bear on this paper. Recommend moving page 13, lines 4-8 up to the introduction.

AC2. Done. We moved page 13, line 4-8 to the introduction in Page 3 line 9-10.

R 3. Page 5, line 13: Here and elsewhere (e.g. Page 12, line 11): Presumably Crocus is running at a 1-hour timestep, and you are outputting daily. Please clarify.

AC3. Done. Page 6, line 1 : ÂńÂăThe Crocus model updates the snowpack every 15 minutes by interpolating meteorological inputs, but in this study we used daily Crocus outputs (SWE, snow depth, density, etc.) computed at 14:00 local time (19:00 UTC), in agreement with the AMSR-2 pass (Sect. 3.1.1)."

R4. Page 5, line 17: "total of precipitable water". Remove "of"

AC4. Done

R5. Page 6, line 11: "the observations errors were". Grammar doesn't work here. Accepted usage should be "the observation error was" but you could also just change

to "observation" and otherwise keep the same.

AC5. Done Page 6: "The observation error was"

R6. Page 7, line 2: "Database" should be "Data".

AC6. Done

R7. Page 8, line 14: "dense forested" should be "densely forested".

AC7. Done

R 8. Page 8, line 15: The signal is not in this case biased. I don't think you can talk about the T_B observation being biased unless e.g. AMSR-2 is measuring TOA values that are biased compared to true TOA values. Instead, I think you mean that it's contaminated or significantly affected by the forest. Treating the TOA measurement as if it were a measurement of T_{B} just above the snow would result in a biased comparison. Anyway, please revise.

AC8. Done. We rewrote the sentence Page 8 Line 17: " The measured TB signal can be significantly affected by the forest and the signature of the underlying snow is attenuated during the winter period in such densely forested areas. "

R9. Page 9, line 21: Crocus has several options for computing grain size. Please give the details here of how this was done for this study, even if they are already reported in the previous Larue et al. 2018 paper. As the authors know so well, T_B is more sensitive to grain size than to SWE, at least at 37 GHz. So this is a really key part of the paper.

AC9. Done. We added this new sentence, Page 9, new line 14: " In particular, the snow layers are modeled with a set of variables representing the morphological properties of snow grains (shape and size), including the specific surface area (SSA), which is one of the most sensitive variables for snowpack emission simulations. The snow microstructure evolves in time according to semi empirical laws (Vionnet et al., 2012).

Crocus is the only model able to simulate the SSA as a prognostic variable (rather than as a diagnostic variable) by using the formulations of Carmagnola et al. (2014). "

R10. Page 10, lines 11-22. I read this a few times, but am still confused. So once detected, an IL is added at the top of the snowpack. Then on the first timestep with precipitation, it is subsequently buried 4 cm beneath the surface? So e.g. it would exist in the model at the top indefinitely as long as there is no snowfall? Why not just add it 4 cm under the surface from the time it is detected?

AC10. We do not integrate it directly at 4 cm because TBs observations are affected at least during 40h (according to satellite observations) by the formation of an IL at the top of the snowpack. We added the following sentence: "This is a simplified way to take into account the presence of IL, and further studies are needed to dynamically evolve these ILs in the snowpack and the impact on the neighboring layers. This work is particularly complex and no solution was found yet (D'Ambroise et al., 2017), in particular because measurements are difficult to take." This section has been clarified. Page 10, new line 17: " Hence, the IL first added at the surface of the snowpack was moved to 4 cm from the surface as soon as a snowfall was detected with GEM precipitation data or, if not, after five days to take into account the snowpack transformations (percolations, sublimations, etc.). The maximum number of detected IL was fixed at two. When a second IL was detected (IL2), IL2 was added at the surface while the first detected IL (IL1) was left at 4 cm. After the next snowfall (or after five days otherwise), IL1 was moved to 8 cm from the surface and IL2 to 4 cm."

R11. Section 3.3, pages 10-12. Overall, I found the notation and presentation to be confusing enough to be distracting here. I would start out the section with an equation that includes both forest and atmosphere; it is frustrating that it starts with an equation neglecting the atmospheric contribution, given the title of the section. I also find it confusing that the atmospheric contributions are presented in a section entitled "Vegetation contributions." Please revise.

AC11. Done, we moved the explanation of the TB, ATM in the section 3.3.

R12. Page 11, line 12. I believe that "simple" should be "single", correct?

AC12. Done

R 13. Page 12, line 5. What does 0.1 represent? Probably better to define as a symbol, and give the value in the text.

AC13. Done

R 14. Page 12, section 3.3.2. Overall I think that you ought to be able to read the section on soil contributions and know which of the parameters are dependent on frequency, and which are frequency invariant. You'll need to revise 3.3.3 a bit too, I think, to avoid duplicating too many explanations.

AC14. Done. See new section 3.3.3

R 15. Page 12, line 6. What is the definition of r_H in equation 8?

AC15. Done

R 16. Page 12, line 8. Is the "." supposed to represent multiplication? If so, please remove, and just take advantage of implied multiplication, writing e.g. \sigma_s = k \sigma.

AC16. Done

R 17. Page 12, line 9-11. Is \Gamma frequency-dependent?

AC17. yes, we rewrote the sentence

R 18. Page 12, line 12-13. I think I see now that you are using \nu to note frequency-dependent variables, and to distinguish from those that are frequency-invariant. However, it took me a while to work this out. Can you reword this, maybe: "Note that we will often use "\nu" subscript to denote quantities that are dependent on frequency, hereafter."

AC18. Done

R19. Page 12 line 15. Sometimes the process of backing out model parameters is referred to as "calibration" and sometimes as "inversion" in this paper. Later (in the results) it's referred to as "optimizations" (Page 17, line 8-9, e.g.). Please just pick one of those two names and use it at all times, to avoid confusion. Else readers wonder if you are referring to the same thing, or to something they missed somewhere in the paper.

AC19. Done, we adjusted the word 'inversion' everywhere.

R 20. Section 3.3.3, pages 12-13. This section required far too long to parse. I found it to be unnecessarily opaque. If this is the same procedure as Roy et al. 14, I would just say that you used the same procedure as that paper. If not, can I recommend a thorough rewrite? Something like: "We thus have two frequency-dependent parameters (eta_nu, beta_nu), and two frequency-invariant parameters (omega, sigma_s). We perform a sort of two-stage calibration. We permute all possible combinations of the two frequency invariant parameters. Specifically we varied omega from 0.02 to 0.16 in steps of 0.01, and varied sigma_s from 0.01 to 1.1 in steps of 0.05. This yields a total of 300 possible combinations of the frequency invariant parameters. Then, for each possible combination of the frequency-invariant parameters, we performed a calibration of the frequency-dependent parameters, eta and beta, for each frequency; thus a total of 900 frequency-dependent calibrations are performed. Finally, for each possible combination of the frequency-invariant parameters, we compute the total post-calibration Tb_RMSE across all three frequencies. The combination of frequency-invariant parameters resulting in the lowest Tb_RMSE is chosen."

AC20. Done, we rewrote the section page 13.

R21. Page 13, lines 9-12. Is the implication that everything is identical to the previous paper except for the covariance inflation? If so, please make this explicit. If not, then no need to highlight covariance inflation prior to beginning the first subsection, in my

opinion.

AC21. We clarified this section Page 13. In this paper we add a new inflation technique of the covariance matrix. It was not performed in the first paper (not necessary with synthetical observations)

R 22. Page 14, line 14. I think ideally you'd have the observation error be larger than 2 K. It really represents all mis-match between observation and model: i.e. what error is expected if the model in its current form is run with "correct" inputs? Of course, this is only a sort of initial value, since you are using covariance inflation. May want to make that explicit here.

AC22. Done. Page 14, new line 20: " Note that in reality it was probably larger since it represents all mismatches between observations and simulations obtained if the model was run with 'correct' inputs. This observation error cannot be easily estimated (low spatial resolution, representativeness, etc.), but it is only a sort of initial value here, since we used a covariance inflation to adjust it."

R 23. Page 14, line 22. Can you clarify that observation error covariance here is just observation standard deviation squared times the identity matrix?

AC23. the R matrix is the observation standard deviation squared times the identity matrix. We clarified it: Page 15, new line 12: " In this study, we developed a new technique to avoid a degeneracy problem, which consists in the online adjustment of the R matrix (i.e. observation standard deviation squared times the identity matrix) such that the weight of the 25-th selected particle (wekeep) is at least equal to 1/N."

R 24. Page 15, line 25. I don't think it is 15% for CoreH2O for shallow snow. I think the requirement was given in absolute SWE (mm) for shallow snow, and a percentage for deep snow. Please double check.

AC24. Rott et al. 2013 detailed the objectives of the CoreH2O mission for SWE esti-mates, and fixed the wanted accuracy to: 3 cm for SWE < 30cm, and 10% for SWE >

30 cm.

R 25. Page 15, line 21. What do you mean by "punctual"? Please reword.

AC25. We mean data simulated at the station. This sentence was removed in page 6 line 16.

R 26. Page 15, line 32. I believe there are twelve total sites. Please make that explicit. Usually you want >20 for e.g. large-sample statistics to hold, right?

AC26. The sentence was clarified. We have 10 SWE stations followed for 4 winters (2012 to 2016), 1 station followed for 2 winters and 1 station followed for one winter, i.e. 43 winter simulation. The winters were very differents from each others (2015-2016 was the warmest from 15 years, and 2014-2015 the coldest from 6 years). The 43 winter simulations were studied together to better represent the different snowpack observed in Québec.

Note that GMON instruments are expensives and daily SWE data are often sparses. This is the first time that the snowpack in Eastern Canada are studied with so much data. 12 others GMON stations were added in 2018.

R 27. Page 15, line 33. Why do you think the site selections are random? In the Western US mountains (albeit a very different environment), it is assumed that logistics of site selection end up leading to a highly biased spatial distribution. E.g. seeMolotch, N. P., and R. C. Bales (2005), Scaling snow observations from the point to the grid element: Implications for observation network design, Water Resources Research, 41(W11421), doi:10.1029/2005WR004229.

AC27. This part was rewritten: Page 6, line 26. " This is a well-known problem for model validation studies and we assume here that the high number of sites (12 SWE stations, or 43 snowpack simulations) provides a useful assessment of simulations. It is also known that the spatial localization of measurements can lead to some biases (Molotch and Bales, 2005). To diversify its measurements, Hydro-Québec has installed

two SWE sensors in the forest, and not in a clearing as is the usual practice for ease of maintenance. "

R 28. Page 16, line 4. I recommend retitling the first subsection "Results of model calibration".

AC28. Done

R 29. Page 16, line 5-7. This entire first paragraph is methods. It must NOT be in the results section. Please move it to the methods section, probably §3.3.3. Also please see my suggestions for reworking §3.3.3.

AC29. Done. See new section 3.3.3.

R 30. Page 17, line 10. I thought you were not calibrating over the winter? Please clarify. Is this using the optimal parameters you obtained over the summer and combining with the open loop model run? Or are you also calibrating over the winter? Recommend describing the winter error statistics very carefully; to be honest, I think having them in there is not worth the added confusion it brings to the reader. The calibration should really be in the background, here, as it has been done in many previous papers. The focus should be on the assimilation results.

AC30. DoneÂă: Section 3.3 was rewritten to be clearer and to well separate method and results. We removed the column with winter RMSE data in the table.

R 31. Page 17, line 12, and elsewhere. "Pluri-annual" is not common English usage. Please reword.

AC31. We replace dit by 'multi-year'

R 32. Page 21, line 12. What is meant by the 48 kg/mËȨ2 limit? This seems to appear from nowhere, and additionally represents very shallow snow.

AC32. we added the following information, but it was explicitly detailed in sect 2.3.4: "Performance is estimated for SWE up to 48 kg m-2 (about 20 cm of snow depth,

derived from measurements, see Sect. 3.4.2)"

R 33. Page 22-23. Recommend simply removing these sub-section headers. They are fairly clear from context, and the sections are not too long. AC33. We thing that the headers help to clarify the presentation of results. Several data are presented here and it can be hard to follow. Moreover, the section is quite long (2 pages).

R 34. Page 22. I don't think you can claim 0.79>0.78 without doing a rigorous statistical test; they seem basically the same to me. The offset is a definitely change; I would highlight that.

AC34. done. The sentence was rewritten (Page 22, line 19) " Correlation between SWEDA simulations and SWE measurements gives a similar R coefficient to the one obtained with SWECrocus simulations (R = 0.79 and R = 0.78, respectively), but the offset is significantly reduced with SWEDA compared to SWECrocus (offset = 10 kg m-2 and 29 kg m-2, respectively). "

R 35. Page 22, line 25. Recommend giving RPE here instead / in addition, since that's what is being discussed earlier in the paragraph.

AC35. Done: RPE GLOBSNOW = 39.5%

R 36. Page 23, Table 5. Recommend redoing notation. Why are italics used in random places? Why aren't "obs" and "sim" subscript?

AC36. Done, see new table 5.

R 37. Page 24, line 8. Recommend introducing "wet snow" as an issue in the introduction. Add some text maybe on how passive microwave won't give additional information about snow once the snow is wet, but can help correct earlier biases, etc. It is only mentioned once in the methods, and is quite easy to miss.

AC37. We added the following sentence in the introduction (page 3, line 2): " However, the assimilation of PMW must be used with care, and a good understanding of the in-

teractions between the properties and microwave emission of the snowpack is crucial to avoid degrading the SWE estimates. For instance, the assimilation of passive microwave in wet snow conditions can introduce large uncertainties since the presence of liquid water in the snowpack increases TBs, whereas increases in snow grain size decrease the brightness temperature independent of any change in SWE (Klehmet et al., 2013)."

R 38. Page 25, Figure 8. Can you show the SWE_{DA} posterior ensemble spread, as in the other graphs? I think we should see it get larger when liquid is present, which should enrich the discussion in this section.

AC38. Done, a new Fig 8 was recomputed with the spread of ensembles.

R 39. Page 26, line 27. Do you mean "However", instead of "Nevertheless"?

AC39. Yes, we corrected this sentence.

R 40. Page 27, Figure 9. Is this for posterior or open loop?

AC40. This is before data assimilation as written in the caption.

---

## Author Comment (AC2) · 10 Sep 2018

General comments of the reviewer:

RC1. This paper evaluates the assimilation of AMSR-2 brightness temperature observations at 11 GHz, 19GHz and 37GHz into the Crocus/DMRT-ML models to analyse snow water equivalent. Results are evaluated against in situ data obtained from 12 sites representing different land cover types in Québec, Eastern Canada. This study is very relevant for the scientific community as assimilating radiances to analyse snow conditions in physical snowpack models is of high interest for hydrology and numerical weather prediction applications. The paper shows promising results for moderate veg-

etation cover and the method opens larger scale applications possibilities. The text has a number of language issues and grammar mistakes, some of which are listed below. The manuscript should be checked by a native English speaking colleague.

AC1: The paper was checked by a native English speaking colleague.

RC2. However the paper is very well organised, results are presented with appropriate figures and tables, and very interesting discussion and conclusion sections are provided. I suggest this paper to be accepted for publication in HESS after the comments below are accounted for.

Page 9 lines 11-12: " To generate a three hourly-continuous meteorological forcing database for running Crocus, successive GEM forecasts were taken from the +09 forecast hour to the +18 forecast hour provided at the 00 and 12 UTC analysis time of each day." It is not clear to me what it means. Did the authors take forecasts at 00 UTC steps covering 09UTC to 12UTC and at 12UTC steps covering 18UTC to 21 UTC? Does it match the AMSR2 pass at 1pm local time? The authors should clarify in the text and also it would be clearer to use UTC everywhere, also when describing the AMSR data both UTC and local time could be provided (page 5 section 2.2 and page 7 section 3.1.1).

AC2: We deleted this part since it was already introduced in the previous section, Page 5, line 20: 'The three hourly-continuous atmospheric forcing database provided by the Global Environmental Multiscale weather prediction model (referred to as 'GEM'; Coté et al., 1998) was used to drive the multi-layer Crocus snowpack model (described in Sect. 3.2.1).' This section 3.2.1 focuses on the coupling Crocus/DMRT-ML, and the GEM inputs used are well detailed in Larue et al., 2018 (section 2.2): 'It provides forecasts of meteorological variables every 3 hr and can make predictions of up to 48 hr. Forecasts are updated daily at 00 and 12 UTC analysis times. To generate a 3-hourly continuous meteorological forcing database for running Crocus, successive GEM forecasts were taken from the +09 forecast hour to the +18 forecast hour provided

at the 00 and 12 UTC analysis time of each day.'

The GEM forecasts were taken in order to match the 1 pm local time. We replaced hours in UTC everywhere. Section 2.2, page 6, line 1: " The Crocus model updates the snowpack every 15 minutes by interpolating meteorological inputs, but in this study we used daily Crocus outputs (SWE, snow depth, density, etc.) computed at 14:00 local time (19:00 UTC), in agreement with the AMSR-2 pass (Sect. 3.1.1)."

RC3. Page 10 lines 18-20: " Hence, as soon as a snowfall is detected with GEM precipitation data, the IL firstly added on the top of the surface was positioned 4 cm from the surface in the simulated snow profile. The maximum number of detected IL was fixed at two. In this case, the first detected IL was 20 positioned at 8 cm from the surface and the second at 4 cm after a snowfall was detected." please clarify/reformulate this part. From the first sentence the reader understands that the first IL is at 4cm depth, but from the second and third sentences it is indicated that the first one is at 8cm depth and the second at 4cm.

AC3: This section 3.1.2 was rewritten, Page 10, line 12: " In this study, an IL was added on the top of the simulated snowpack if the AMSR-2 PR(11) was above 0.06 (Roy, 2014). This IL was represented as a 1-cm layer with a density of 900 kg m-3 and with snow grain radius set to zero (Roy et al., 2016). The difficulty is to know how to evolve this IL in the snowpack. The Crocus snowpack model has not yet been adapted to integrate the formation of ILs and evolve them in a coherent way (Quéno et al., 2016). Nevertheless, it was shown in Larue et al. (2018) (from field measurements) that an IL of 1 cm located at 4 cm from the surface of the simulated snowpack minimized the bias of DMRT-ML simulations due to the presence of an IL (regardless of its real location in the snow profile). Hence, the IL first added at the surface of the snowpack was moved to 4 cm from the surface as soon as a snowfall was detected with GEM precipitation data or, if not, after five days to take into account the snowpack transformations (percolations, sublimations, etc.). The maximum number of detected IL was fixed at two. When a second IL was detected (IL2), IL2 was added at the surface

while the first detected IL (IL1) was left at 4 cm. After the next snowfall (or after five days otherwise), IL1 was moved to 8 cm from the surface and IL2 to 4 cm. For instance, during winter 2014-2015, one IL was detected at sites 1 and 12 (22 December 2014 and 15 December 2014). At Site 9, two ILs were detected: one on 10 December 2014 and another on 1 January 2015. "

RC4. In section 3.4.2 it would be very useful to include a table with the list of experiments with a short name for each and indicating in the caption the experiment period and sites. So that the reader would have centralised in the table the experiment set-up information. For example experiment names like "DA1_TB19-37", "DA2_TB19-37,TB11-19", "DA3_TB_11,19,37". Using these short names in Section 4 when presenting the results would be much clearer.

AC4: We added a new Table (table 3) with acronyms. We used the new acronyms in section 4 to be clearer.

RC5. Also the first sentence of the section 3.4.2 starts with "In a first step,...", it should be followed by "In a second step," probably page 15, line 17 when introducing the experiment on all sites. It is not a problem to reveal the fact that the best configuration will be DA of TB11,19,37 at this stage, it will only make the paper clearer. Page 18: section 4.2.1, first paragraph: there is no need to repeat in the results section the experiments that were conducted: please remove the first sentence of section 4.2.1.

AC5: Done, these sections were checked to be clearer and the first sentence of section 4.2.1 was removed to avoid repetitions.

RC6. Finally, even the last experiment, with free snow stickiness and forest parameter, presented in Section 5, should be described in Section 3.4.2 and included in the experiment list table. It is surprising for the reader to be informed in Section 5 that another experiment was conducted. In other words, the paper should not necessarily follow the chronology of the research developments. It should present the experiments and the results/discussion without holding new experiment description for the results and

discussion sections.

AC6: Done, we moved it in the result section, and introduced it in the method as the 'experiment C'. see new section 3.4.2.

RC7. Page 20 line 6 and Figure 6: The way it is formulated page 20 and in the caption, it is not straightforward to understand the meaning of "SWE ensemble obtained with the DA of the three frequencies (referred to as 'SWEDA')". The reader may wonder why this one is called SWEDA whereas the DA of TB 11, 19 37 GHZ had no specific name. Please use experiment names and provide a table in section 3 (see comment above).

AC7: Done, we used experiment names introduced in new table 3 (section 3).

Minor comments of the reviewer:

RC8. table 1 Caption: replace: "Characteristics of the nivometric stations: SWE (in kg m-2) data, Latitude (Lat.),..." by "Characteristics of the nivometric SWE stations: Site number, Latitude (Lat.),"

AC8. Done

RC9. Table1 caption: GEM is used here but only defined later in section 2.2. So, define it on its first occurence in Table 1's caption.

AC9. Done

RC10. Page 5, line 11: remove "further"

AC10. Done

RC11. Page 5 line 13: replace "1pm" by "1pm local time"

AC11. Done

RC12. Page 5 last paragraph: make sure tenses are consistent: line 12: "Crocus computes" and line 13-14: "The DMRT-ML ... was used to "

AC12. Done

RC13. Page 6 line 13: ".." -> "."

AC13. Done

RC14. Page 8, table 2, caption: clarify if the winter period January to March as indicated in the caption, or if it is 1 January to 1 March as indicated in the text page 7 line 18.

AC14. Done

RC15. Page 8 line 17 "formulations in" -> "formulations of"

AC15. Done

RC16. Page 10 line 24: reformulate by something like: "The PMW brightness temperature (TB) emitted at the scale of the AMSR-2 product can be written as (2) for each grid cell as"

AC16. Done

RC17. Page 11 line 15: "the expression of TB TOA in boreal areas was described by the Eq. (2)" for consistency please update Eq 2 (page 10) by replacing "TB=" by "TB TOA="

AC17. Done

RC18. Page 12 line 3: remove "throughout the year"

AC18. Done

RC19. Page 12 line 7: "In the Eq.7"–> "In Eq. 7"

AC19. Done

RC20. Page 12 lines 17-20: update the text to ensure tenses consistency. For example lines 17-18: "Forest parameters (!, _) depend on the forest characteristics, such as the

biomass and the structure of the canopy for each site. To take into account the temporal variations of these caracteristics, the forest parameters were linked to the LAI." can be replaced by: "Forest parameters (!, _) depend on the forest characteristics, such as the biomass and the structure of the canopy for each site. They also depend on LAI which allows to account for the seasonal cycle in the forest emission". Also check the rest of the paragraph.

AC20. Done

RC21. Page 12 line 31: replace "value couple" by "set of values", replace "(considered constant in frequency)" by ", defined at each frequency (11 GHz, 19 GHz and 37 GHz)", and replace "for each frequency (at 11, 19 and 37 GHz) in V-pol" by "at V-pol".

AC21. Done

RC22. Page 12 line 32 / page 13 line 1: This sentence is not clear, it should be reformulated. Do you mean that the parameters were optimised also at H-pol or that the V-pol set of parameters were tested at H-pol?

AC22. DoneÂǎ; We used V-pol only since H-pol is very sensitive to the stratigraphy of the snowpack and to the presence of ILS. Section 3.4.2: "We used V-pol TB because H-pol TB is more sensitive to the stratigraphy of the snowpack and to the presence of ILs (Mätzler, 1987). ". Moreover, this section was rewritten to be clearer, Page 12, new line 29: ' the two frequency-dependent parameters ($\eta\nu$, $\beta\nu$) and two frequency-invariant parameters ($\omega$, $\sigma$s) were inverted with a two-stage calibration by permuting all possible combinations of the two frequency invariant parameters. Specifically, $\omega$ values varied from 0.02 to 0.16 in steps of 0.01, and $\sigma$s varied from 0.01 to 1.1 in steps of 0.05. This yields a total of 300 possible combinations of the frequency invariant parameters. Then, for each possible combination of the frequency-invariant parameters, a calibration of the frequency-dependent parameters, $\eta\nu$ and $\beta\nu$, was performed for each frequency. A total of 900 frequency-dependent calibrations were thus computed. Finally, for each possible combination of the frequency-invariant parameters, the total post-calibration

TB RMSE across all three frequencies was computed. The combination of frequency-invariant parameters resulting in the lowest TB RMSE was chosen.'

RC23. Page 13 lines 9-12: the 3 sentences on observation errors and ensemble inflation technics should be removed because they are not well formulated and the description given on the next page (page 14) is very clear.

AC23. Done

RC24. Page 14, lines 19-21: This sentence repeats lines 12-13: " Hence, to avoid a 20 degeneracy problem, the weight of the 25-th selected particle (wekeep) must always be larger or equal to the inverse of the ensemble size (N=150)." Please update the text to avoid repeating sentences.

AC24. Done

RC25. Page 15, first paragraph, last sentence: this statement should be placed earlier in the paragraph, before the three experiments are described.

AC25. Done

RC26. Page 14 line 30 and page 15 line 8: The information on the DA experiments length and period should not be spread in the test. It should be clearly stated once.

AC26. Done. This section was rewritten.

RC27. Page 16 line 5: remove "(constant in frequency, Sect. 3.3.3)"

AC27. Done

RC28. Page 16, section 4.1: update the text to use consistent tenses.

AC28. Done, we checked all the tenses in the text.

RC29. Page 18 line 19: replace "according to the studied period" by "for the studied period"

AC29. Done

RC30. Page 27 lines 14-15: replace "explain" by "explains" and "up to" by "larger than".

AC30. Done

---

## Author Comment (AC3) · 10 Sep 2018

Dear Editor-in-chief,

Please find attached to this letter the revisions to this manuscript. We would like to thank the reviewers for their detailed comments which were useful in producing an improved manuscript. We carefully responded to all the questions and comments. As requested by the reviewers, the manuscript was checked by a native English professional reviewer to correct language issues and grammar mistakes. We put in advance a tiny mistake in the assimilation algorithm. This was not significant, we, nevertheless, recalculate all the simulations. As expected, the difference compared to previous re-

sults are not important, but we present all the new results in this revised version, as this is more rigorous. It doesn't change the conclusion of the paper. Note that since the first submission of this paper, our paper previously submitted and explaining in details the assimilation scheme used in this paper is now published in Water Resources Research (Larue, F., Royer, A., De SeÌÁve, D., Roy, A., Picard, G., Vionnet, V., & Cosme, E.: Simulation and assimilation of passive microwave data using a snowpack model coupled to a calibrated radiative transfer model over northeastern Canada. Water Resources Research, 54, 4823-4848, https://doi.org/10.1029/ 2017WR022132, 2018). Moreover, a strong effort has been made to facilitate a deeper understanding of this paper (flow chart modified, methodology better described. . .). A new table was added to describe all acronyms and experiments. We think that this new version is significantly improved compared the previous submission.

Please, do not hesitate to contact me if you have any further questions or comments, or if anything is missing from the submission of the revised manuscript.

Regards,

Fanny LARUE

Please also note the supplement to this comment:
https://www.hydrol-earth-syst-sci-discuss.net/hess-2018-95/hess-2018-95-AC3-supplement.pdf

**Supplement:**

[revised manuscript text omitted]

---

## Author Comment (AC4) · 11 Sep 2018

[revised manuscript text omitted]
 | Bias | RPE | $\overline{\text{SWE}_{sim}}$ | RMSE | Bias | RPE | $\overline{\text{SWE}_{sim}}$ |
| | | (kg.m$^{-2}$) | (kg.m$^{-2}$) | (%) | | (kg.m$^{-2}$) | (kg.m$^{-2}$) | (%) | |

[revised manuscript text omitted]